# Prospective study to explore changes in quality of care and perinatal outcomes after implementation of perinatal death audit in Uganda

Victoria Nakibuuka Kirabira,[1] Mamuda Aminu [ID] ,[2,3] Juan Emmanuel Dewez,[2] Romano Byaruhanga,[1] Pius Okong,[1] Nynke van den Broek[2]

[1]Paediatrics Department, Nsambya Hospital, Kampala, Uganda
[2]Centre for Maternal and Newborn Health, Liverpool School of Tropical Medicine, Liverpool, UK
[3]Department of International Public Health (DIPH), Liverpool School of Tropical Medicine, Liverpool, UK

**Correspondence to**
Dr Mamuda Aminu;
mamuda.aminu@lstmed.ac.uk

## ABSTRACT

**Objective** To assess the effects of perinatal death (PND) audit on perinatal outcomes in a tertiary hospital in Kampala.

**Design** Interrupted time series (ITS) analysis.

**Setting** Nsambya Hospital, Uganda.

**Participants** Live births and stillbirths.

**Interventions** PND audit.

**Primary and secondary outcome measures** Primary outcomes: perinatal mortality rate, stillbirth rate, early neonatal mortality rate. Secondary outcomes: case fatality rates (CFR) for asphyxia, complications of prematurity and neonatal sepsis.

**Results** 526 PNDs were audited: 142 (27.0%) fresh stillbirths, 125 (23.8%) macerated stillbirths and 259 (49.2%) early neonatal deaths. The ITS analysis showed a decrease in perinatal death (PND) rates without the introduction of PND audits (incidence risk ratio (IRR) (95% CI) for time=0.94, p<0.001), but an increase in PND (IRR (95% CI)=1.17 (1.0 to −1.34), p=0.0021) following the intervention. However, when overdispersion was included in the model, there were no statistically significant differences in PND with or without the intervention (p=0.06 and p=0.44, respectively). Stillbirth rates exhibited a similar pattern. By contrast, early neonatal death rates showed an overall upward trend without the intervention (IRR (95% CI)=1.09 (1.01 to 1.17), p=0.01), but a decrease following the introduction of the PND audits (IRR (95% CI)=0.35 (0.22 to 0.56), p<0.001), when overdispersion was included. The CFR for prematurity showed a downward trend over time (IRR (95% CI)=0.94 (0.88 to 0.99), p=0.04) but not for the intervention. With regards CFRs for intrapartum-related hypoxia or infection, no statistically significant effect was detected for either time or the intervention.

**Conclusion** The introduction of PND audit showed no statistically significant effect on perinatal mortality or stillbirth rate, but a significant decrease in early neonatal mortality rate. No effect was detected on CFRs for prematurity, intrapartum-related hypoxia or infections. These findings should encourage more research to assess the effectiveness of PND reviews on perinatal deaths in general, but also on stillbirths and neonatal deaths in particular, in low-resource settings.

### Strengths and limitations of this study

► This study is one of the few studies assessing the effect of perinatal death audits in a resource-limited setting.
► The inherent limitation of the methods used to assess the cause of death was the dependence on hospital records, which were often incomplete, inaccurate or both.
► The interrupted time series (ITS) design was useful to assess the effect of the intervention, but only two time points of data were documented before the introduction of the audit, which could have influenced the effect on outcomes of interest.
► Additional studies, such as ITS studies with longer preintervention time points of data collection, stepped-wedge randomised controlled trials, and perhaps postmortem autopsies, must confirm the findings of this study.

## BACKGROUND

Currently, about 2.7 million neonatal deaths and 2.6 million stillbirths occur worldwide every year. Of these, the majority occur in low-income and middle-income countries, and more than half (55%) of the global stillbirths occur in sub-Saharan Africa and South Asia.[1] Uganda's perinatal mortality is estimated to be 36 deaths per 1000 births and remains unacceptably high. Almost 50% of these are fresh stillbirths.[2] The neonatal mortality rate (21 per 1000 live births) in Uganda is also high, contributing 50% of the infant mortality rate.[3] It is crucial that interventions are put in place to reduce preventable neonatal deaths and stillbirths.

Perinatal death audit (or review) can be used to identify areas of care which can be strengthened to improve perinatal outcomes.[4] It involves reviewing cases of mortality with a view to identifying avoidable factors and implementing changes to prevent future deaths. A population-based cohort study of

943 perinatal mortalities that occurred in 90 hospitals in the Netherlands reported a decrease in substandard care (from 10% to 5%) and perinatal mortality (from 2.3 to 2.0/1000 births; p<0.00001) after the introduction of perinatal death audit.[5]

Similarly, in a systematic review focusing on low-income and middle-income countries, a meta-analysis of seven before-and-after studies reported a 30% (95% CI 21% to 38%) reduction in perinatal mortality after introduction of perinatal death audit in healthcare facilities.[6] However, the authors considered most studies included in the review as of low quality. Thus, there is a paucity of data from low-resource settings on the use of and effectiveness of perinatal death audit to improve the quality of care provided to mothers and their babies. The systematic review recommended more research in key areas, including exploring the effect of facility-based perinatal death audit on changing the quality of care.[6]

The aim of this study is to assess the changes in perinatal outcomes observed after the introduction of perinatal death audit in a tertiary hospital in Uganda and to describe the changes in healthcare provided to mother and babies resulting from the perinatal death audits.

## METHODS
### Study setting
This study was conducted at Nsambya Hospital, a private not-for-profit tertiary hospital in Kampala, Uganda, which serves as a referral centre for Makindye Health subdistrict in Kampala. The facility receives 7000 deliveries per year and provides 24 hours comprehensive emergency obstetric and newborn care services, with a staff of 6 obstetricians, 18 resident doctors, 5 intern doctors and 40 midwives.

The newborn care unit admits 2500 babies annually and has a paediatrician/neonatologist, two residents and up to four midwives on duty per shift. It has 40 beds, including 10 neonatal intensive beds, and provides essential newborn care including; oxygen therapy, surfactant and low-cost bubble continuous positive airway pressure (CPAP), kangaroo mother care (KMC) and phototherapy.

### Data collection
Perinatal death audit using locally developed tools and guidelines started in 2008 and was led by a multidisciplinary audit team comprising of obstetricians, paediatricians, resident doctors, midwives and administrators. Team meetings were held weekly, chaired by the most senior obstetrician or paediatrician. Perinatal death was defined as 'death of a fetus greater than a birth weight of 1000 g if measured, or a gestational age of at least 28 completed weeks or more, or a body length (crown-heel) of at least 35 cm or more'[7] and included all stillbirths and deaths within the first week of life (early neonatal deaths). The number of perinatal deaths was documented daily using clinical handover reports and registers in the labour ward and neonatal special care unit. Due to the large

numbers of deaths, one-quarter of the cases (every fourth case) were selected for audit from each of the following categories: fresh stillbirths, macerated stillbirths and early neonatal deaths.

The number of admissions in the newborn unit, reasons for admission, death and case fatalities rates for intrapartum-related hypoxia, infection and prematurity were collected for the period before the perinatal audit was introduced (2006–2007) and subsequently for the period after commencement of the perinatal death audit (2008–2015). The data were obtained and aggregated monthly through a retrospective analysis of registers and patient records. Numbers of live and total births were also documented to enable rates calculation.

### Intervention: the audit process
For each case selected for review, an audit form was completed by a resident doctor, who also provided a narrative summary for each case. Sources of information included: clinical handover books, maternity registers, newborn admission register and clinical notes for mothers and newborns. Each case was then presented to the rest of the audit team for detailed discussion. The information presented to the team allowed reviewing the cause of death and assigning a more precise cause of death by consensus. Care provided was judged against existing local protocols and national guidelines. No clinical autopsies were performed due to lack of a full-time pathologist at the hospital.

The quality of care provided at the hospital was also graded by consensus and classified into four categories:
► Optimal care generally accepted practice and all standards of care were followed.
► Suboptimal care but different management would have made no difference to the outcome (probably acceptable).
► Suboptimal care where different management might have made a difference to the outcome—an avoidable factor of uncertain clarity or influence on the outcome (probably suboptimal).
► Different management would reasonably be expected to have made a difference to the outcome. A clearly avoidable factor implying that the adverse outcome could have been prevented) (suboptimal).

The overall grading of the care provided to the mother and baby, the identified specific areas of suboptimal care and the recommendations for improvement were documented. This grading included referred mothers as well, whereas there were delays from the referring site, suboptimal care can also occur in tertiary hospital as well, such as delay in performing a caesarean section.

The summary of each case included a list of the actions to be taken. In each case, a specific person was appointed to lead the implementation of the relevant actions. During monthly meetings, implementation progress was compared with the action plan by the wider audit team. A number of interventions, such as neonatal resuscitation training, the introduction of KMC, building of a new

maternity theatre, were thus introduced gradually over the years to address the gaps in care. Whereas maternal death reviews had been introduced before the perinatal death audits, they emphasised more care to the mother than the baby.

### Data analysis

SPSS (IBM, V.22) was used for descriptive analyses. Stillbirth and perinatal mortality rates were expressed per 1000 total births, while the early neonatal mortality rate was expressed per 1000 live births. Case fatality rate (CFR) was defined as the proportion of deaths among cases diagnosed with the disease. Statistical significance was calculated at 95% CI or p value of <0.05.

Data for the six primary and secondary outcomes were each analysed using interrupted time series (ITS) models in Stata, following Cochrane checklist for assessing the quality of ITS studies as a guide. Poisson regression models of counts were fitted with covariates for intervention and year, and the relevant population size (log-transformed) used as the offset. As proposed by Bernal et al,[8] overdispersion was considered (by using an additional option: scale (x2)) since it often arises in the context of count data. Autocorrelation was also inspected using plots. The 5% level was used to determine statistical significance.

Since low birth weight and prematurity are both associated with a higher risk of perinatal mortality, a one-way between-groups analysis of variance (ANOVA) was conducted to explore changes in birth weight and gestational age over time.

A narrative summary description was used to report actions agreed and implemented following an audit. These were documented chronologically as derived from minutes of the audit meetings.

### Patient and public involvement

Individual patient consent was not applicable as the research team extracted anonymised data from case notes. Patients and the public were not involved in this study.

### RESULTS

#### Demographic and clinical characteristics

A total of 58 997 births and 2616 perinatal deaths occurred between 2008 and 2015 (online supplementary table 1). Of these, 603 (23%) perinatal deaths were selected for audit. After review, 526 were found to meet the definition of perinatal mortality: 142 (27.0%) fresh stillbirths, 125 (23.8%) macerated stillbirths and 259 (49.2%) early neonatal deaths (table 1).

The majority of cases came to the hospital directly, with 17% referrals. Mothers had a mean age of 27.3 years (SD: 5.1); median parity of 2 (SD of 1.6). Just over one-third of fresh stillbirths and early neonatal deaths were delivered by caesarean section. The mean birth weight of babies was 2610 g (SD: 969.4). Between 23.2% and 39.4% were born prematurely (table 1).

### Cause of death

Overall, intrapartum-related hypoxia was the leading cause of perinatal mortality in this setting, accounting for 35.4% of all cases (figure 1 and online supplementary table 2). This is followed by respiratory distress syndrome (8.9%) and meconium aspiration pneumonia (7.6%). Cause of death in 43.2% could not be ascertained: 15.4% for neonatal deaths, 49.2% for fresh stillbirths and 88.0% for macerated stillbirths. However, the proportion of deaths for which cause of death could not be established ('cause unknown') decreased from 34.1% (overall) in 2008 to 2.7% in 2013.

### Trend in perinatal mortality rate

The ITS analysis showed that, without modelling overdispersion, there would have been a downward trend (a decrease) in perinatal death rates (incidence risk ratio (IRR) (95% CI) for time=0.94, p<0.001), while a statistically significant increase (IRR (95% CI)=1.17 (1.0,–1.34), p=0.0021) was observed following the introduction of perinatal death audits. However, when overdispersion was included in the model, both effects were not statistically significant (table 2, figure 2 and online supplementary table 3).

Stillbirth rates exhibited a similar pattern. By contrast, early neonatal death rates showed an overall upward trend (IRR (95% CI)=1.09 (1.01–1.17), p=0.01) but a a decrease following the introduction of the perinatal death audits (IRR (95% CI)=0.35 (0.22–0.56), p<0.001) when overdispersion was included in the model (table 2 and figure 2). The ANOVA to explore changes in birth weight and gestational age over time showed no statistically significant change in birth weight over time (p=0.872), and no change in gestational age at birth over time (p=0.219).

The CFR for prematurity showed a downward trend over time (IRR (95% CI)=0.94 (0.88–0.99), p=0.04) but a downward trend was not observed with the intervention. With regards to the CFRs for intrapartum-related hypoxia and infections, no statistically significant effect was detected with or without the intervention (table 2, figure 2 and online supplementary table 3).

For each outcome considered, the autocorrelation and partial correlation plots provided no evidence that there was autocorrelation not accounted for in the models used.

### Quality improvement

More than half of the cases (53%) were deemed to have had optimal care. The rest had probably acceptable care (21.7%), probably suboptimal care (16.6%) or suboptimal care (8.7%) (online supplementary table 4). Among the different types of mortalities, suboptimal care was most common among early neonatal deaths (11.8%), as compared with fresh stillbirth (6.8%) and macerated stillbirth (5.0%).

### DISCUSSION

#### Main findings

This study assessed the effect of perinatal death audit to improve the quality of perinatal care and perinatal

**Table 1** Demographic and clinical characteristics

| Characteristics | Fresh stillbirth n=125 (%) | Macerated stillbirth n=142 (%) | Early neonatal death n=259 (%) | Total n=526 (%) |
|---|---|---|---|---|
| **Referral** | | | | |
| From other healthcare facility | 32 (25.6) | 15 (10.6) | 37 (14.3) | 84 (16.0) |
| From traditional birth attendant | 1 (0.8) | 1 (0.7) | 1 (0.4) | 3 (0.6) |
| **Parity** | | | | |
| 1 | 35 (28.0) | 48 (33.8) | 129 (49.8) | 212 (40.3) |
| 2–4 | 64 (51.2) | 67 (47.2) | 84 (32.4) | 215 (40.9) |
| 5 or more | 10 (8.0) | 15 (10.6) | 18 (6.9) | 43 (8.2) |
| **Antenatal care** | | | | |
| At least 1 ANC visit | 107 (85.6) | 129 (90.8) | 234 (90.3) | 470 (89.4) |
| 4 or more ANC visits | 38 (30.4) | 47 (33.1) | 98 (37.8) | 183 (34.8) |
| **Type of pregnancy** | | | | |
| Single pregnancy | 113 (90.4) | 131 (92.3) | 227 (87.6) | 471 (89.5) |
| Multiple pregnancy | 2 (1.6) | 5 (3.5) | 14 (5.4) | 21 (4.0) |
| **Mode of delivery** | | | | |
| Spontaneous vaginal delivery | 67 (53.6) | 123 (86.6) | 143 (55.2) | 333 (63.3) |
| Assisted breech delivery | 6 (4.8) | 2 (1.4) | 10 (3.9) | 18 (3.4) |
| Instrumental delivery | 4 (3.2) | 1 (0.7) | 3 (1.2) | 8 (1.5) |
| Caesarean section | 45 (36.0) | 10 (7.0) | 92 (35.5) | 147 (27.9) |
| **Gestational age at delivery** | | | | |
| <37 weeks | 29 (23.2) | 56 (39.4) | 74 (28.6) | 159 (30.2) |
| 37–41 weeks | 60 (48.0) | 57 (40.1) | 132 (51.0) | 249 (47.3) |
| >41 weeks | 5 (4.0) | 4 (2.8) | 9 (3.5) | 18 (3.4) |
| **Baby's sex** | | | | |
| Male | 68 (54.4) | 80 (56.3) | 146 (56.4) | 294 (55.9) |
| Female | 52 (41.6) | 57 (40.1) | 99 (38.2) | 208 (39.5) |
| **Birth weight** | | | | |
| <1000 g | 1 (0.8) | 16 (11.3) | 9 (3.5) | 26 (4.9) |
| 1000–2499 g | 30 (24.0) | 52 (36.6) | 81 (31.3) | 163 (31.0) |
| 2500–4000 g | 77 (61.6) | 61 (43.0) | 149 (57.5) | 287 (54.6) |
| >4000 g | 3 (2.4) | 7 (4.9) | 6 (2.3) | 16 (3.0) |

ANC, antenatal care.

outcomes in a tertiary hospital in Uganda over a period of 10 years.

Intrapartum-related hypoxia was the leading cause of perinatal mortality (accounting for 35.4%), followed by respiratory distress syndrome (8.9%) and meconium aspiration pneumonia (7.6%). Cause of death was unknown in 43.2% of cases. The proportion of deaths with unknown cause decreased from 34.1% (overall) in 2008 to 2.7% in 2013.

There was no effect of perinatal death audit on perinatal mortality or stillbirth rates. However, the intervention led to a decrease in neonatal mortality rate (IRR (95% CI)=0.35 (0.22–0.56), p<0.001)). It was noted that the high numbers of stillbirth inflated the numbers of perinatal death. Thus, even though the audit made a difference to early neonatal deaths, this difference is neutralised when combined with stillbirths.

The CFR for prematurity showed a downward trend over time (IRR (95% CI)=0.94 (0.88 to 0.99), p=0.04), but this was not related to the perinatal death audits. The CFRs for intrapartum-related hypoxia and infections were not affected by the intervention.

More than half of the cases (53%) were deemed to have had optimal care, with the rest of the cases having varying levels of acceptability. Early neonatal deaths had the highest proportion of cases with suboptimal care (11.8%). CFRs decreased between 2006 and 2015 for intrapartum-related hypoxia (21.3%–15.5%) and for complications of

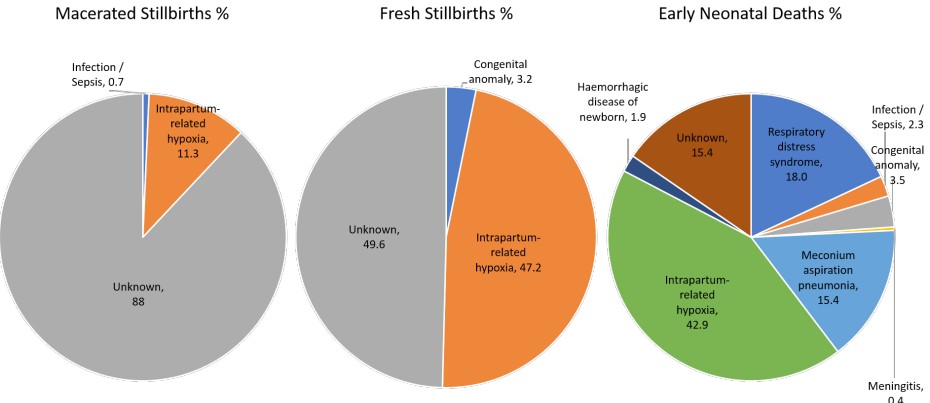

**Figure 1** Cause of death for macerated stillbirths (n=142), fresh stillbirths (n=125) and early neonatal deaths (n=261).

prematurity (26.4% to 11.1%). However, CFRs for infection increased from 1.9% in 2006 to 5.7% in 2015.

### Quality improvement

In 2007 and 2008, the perinatal death review team observed many cases of delay between the decision for, and the actual conduct of, caesarean Section, which was associated with high numbers of intrapartum-related hypoxia. To reduce the time taken to prepare mothers for the procedure, the team recommended the involvement of doctors in the preparation. These ensured mothers were catheterised and blood samples were taken for grouping and cross-matching at the time of decision. During the same period, a new dedicated operating theatre was built close to the maternity ward.

In 2008, two major factors contributing to perinatal mortality were identified to have recurred consistently: poor neonatal resuscitation and poor use of the partograph. Both of these were thought to contribute to perinatal death due to intrapartum-related hypoxia. To tackle the problem of poor resuscitation skills, maternity staff were retrained in resuscitation and mentored by the most senior staff. In addition, senior staff were involved earlier in deliveries among high-risk patients. The hospital also acquired more resuscitation equipment, including

dedicated Ambu bags, masks and Resuscitaire for the labour ward and theatre. Midwives were retrained in the use of the partograph. Human resource challenges were addressed with more midwives assigned to the maternity unit.

Training in neonatal resuscitation was done on a 3 monthly basis. In 2007, asphyxiated babies were only given 10% dextrose. In subsequent years after the training, the care was expanded to include providing warmth, stimulation and ventilation with Ambu bag and provision of oxygen. This was implemented both in the labour wards and theatre.

In 2009, the CFR for complications of prematurity was observed to begin to rise (figure 2). This prompted action to address the problem of hypothermia, which was observed to be related to many of the cases. The team introduced KMC for stable preterm babies. They also recommended the administration of dexamethasone to mothers at risk of preterm birth, and use of radiant warmers in the theatre, labour room and the newborn unit to improve thermal control. For babies with respiratory distress, CPAP was introduced, and rescue surfactant was made available for use in 2014.

| Table 2 | Interrupted time series analysis for selected perinatal outcomes | | | | |
|---|---|---|---|---|---|
| | **With overdispersion** | | | | |
| | **Intervention** | | **Time** | | |
| Outcome | **IRR (95% CI)** | **P value** | **IRR (95% CI)** | **P value** | |
| Perinatal death rate | 1.17 (0.79 to 1.74) | 0.44 | 0.94 (0.89 to 1.00) | 0.06 | |
| Stillbirth rate | 1.31 (0.84 to 2.06) | 0.23 | 0.93 (0.87 to 1.00) | **0.039** | |
| Early neonatal death rate | 0.35 (0.22 to 0.56) | **<0.001** | 1.09 (1.01 to 1.17) | **0.01** | |
| CFR for intrapartum-related hypoxia | 0.77 (0.57 to 1.04) | 0.09 | 0.98 (0.93 to 1.03) | 0.40 | |
| CFR for prematurity | 1.01 (0.64 to 1.59) | 0.96 | 0.94 (0.88 to 0.99) | **0.04** | |
| CFR for infections | 2.25 (0.17 to 28.9) | 0.54 | 1.04 (0.78 to 1.39) | 0.77 | |

Bold values indicate "statistically significant"
CFR, case fatality rate; IRR, incidence risk ratio.

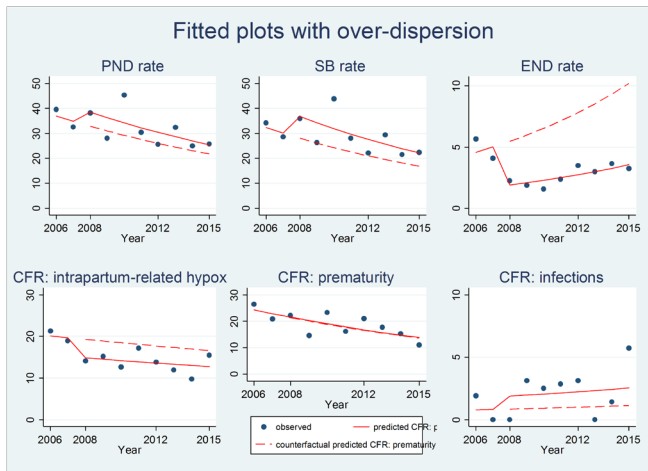

**Figure 2** Interrupted time series with overdispersion. CFR, case fatality rate; END, early neonatal death; PND, perinatal death; SB, stillbirth.

In 2012, light-emitting diode phototherapy was introduced for the management of babies with neonatal jaundice. No additional specific actions to improve the management of neonatal infections were documented as this was not considered a major cause of mortality when compared with intrapartum-related hypoxia and complications of prematurity. However, all interventions were observed to be sustained throughout the period under study. This study assessed the introduction of perinatal audits in a low-resource setting. There are few studies exploring the use of perinatal death audit to improve care in low-resource settings. In a similar study of 305 perinatal deaths in Tanzania, suboptimal care was identified in 80% of the cases. However, there was no report of changes implemented or on the effect of the changes.[9]

### Strengths and limitations
This study is one of the few studies assessing the effect of perinatal death audits in a resource-limited setting, over a period of ten years. While it is difficult to attribute all changes in perinatal outcomes to perinatal death audit alone, the process has helped to identify areas of healthcare that need support for improvement in outcomes.

The results should be interpreted in light of the limitations. The attribution of cause of death was based on hospital records, which were often incomplete, inaccurate or both. In terms of the ITS design, there were only two time points of data before the introduction of the audits, which limits the capacity of the analysis to examine evidence of a change in slope with the adoption of the intervention. A third time point, as suggested in the Cochrane standards for ITS, would have addressed this issue.

### Implications for future practice and research
#### Problems identified through audit
Perinatal death audit was integrated as part of routine care to help identify deficiencies in the clinical care pathway and to formulate priority actions to ensure improved quality of care and reduced loss of life. Poor use of the partograph and fetal heart monitoring, delay in performing caesarean sections when these were indicated, poor neonatal resuscitation skills and poor management of preterm babies, were identified as areas requiring improvement in the quality of care. In addition, there was evidence of significant delays incurred when a referral from a lower level healthcare facility occurred. These findings are similar to a study from Tanzania where 47% of perinatal deaths audited had poor fetal heart monitoring, and 23% had delays in performing caesarean section.[9 10]

The interventions for improvement included: training in the use of the partograph, fetal heart monitoring, neonatal resuscitation, measures to reduce delay in performing a caesarean section by involving doctors in the timely preparation of mothers before transfer to the theatre and the building of a separate maternity theatre. The implementation of these interventions was monitored during the weekly audit meetings. Training in neonatal resuscitation has been conducted in Malaysia and Malawi with similar success in the acquisition of skills.[11 12]

### Interventions for prematurity-related complications
The interventions implemented included: reduction of hypothermia through the provision of radiant warmers and incubators for the critically ill babies in the labour wards, theatre and the newborn unit. KMC and immediate skin-to-skin care at birth were initiated in 2009 and 2012, respectively. The use of skin-to-skin care was noted to improve from 40% to 80% of live births after repeated training. KMC has been adopted and is used for all stable low birth weight babies in the newborn unit. A Cochrane systematic review has established that KMC is associated with reduced risk of neonatal mortality (by 51%), nosocomial infection or sepsis (by 58%) and hypothermia (by 77%).[13] The interventions for reduction of respiratory distress included: use of antenatal steroids, bubble CPAP and use of rescue surfactant. Antenatal steroids, bubble CPAP and rescue surfactant have been found to have low coverage in low-resource settings despite their potential to reduce mortality by 48%, 31% and 32%, respectively.[14–16] The majority (98%) of the preterm babies who needed it were able to pay for it and actually received it. This is a private not for profit hospital, majority of the patients that are admitted are able to afford the rates offered by the hospital. Low-cost surfactant was obtained from South Africa/India which cost about 200 dollars per vial. Surfactant is a product that is paid for in our hospital just like other commodities like antibiotics or caesarean section. The cost of surfactant is less than an operation like caesarean section.

### Intrapartum-related hypoxia and prematurity-related deaths
In this study, the CFR for intrapartum-related hypoxia was reduced from 21.3% to 15.5%. For stillbirth, intrapartum-related hypoxia has been reported to account for between 3.1% and 25% of all cases of stillbirths in low-income and

middle-income countries.[17] The change may be attributable to the improvement in resuscitation practices, following training and provision of equipment. Training in neonatal resuscitation has been reported to reduce deaths in babies with intrapartum-related hypoxia by 30% and early neonatal deaths by 38%.[18 19] An analysis of seven facility-based studies estimated that training in neonatal resuscitation reduced the neonatal mortality rate from 43% to 17%. However, a study in Malawi reported that resuscitation training did not reduce mortality.[12] In Malawi, factors such as availability and replacement of resuscitation equipment and mentorship after training were not part of the improvement package. Additionally, in our study, the training was repeated on a 3 monthly basis and continuous feedback and mentoring were provided which helped ensure continuous improvement in quality. Although these interventions had already been known to work, audit acted as a precursor to implement them.

It is noteworthy that intrapartum-related hypoxia was documented as a cause of death even for macerated stillbirth. Since maceration could set in only 6 hours postmortem,[20] it is not uncommon in low-resource settings to find a macerated intrapartum death due to delays in an intervention during labour. It could also be because healthcare providers often misclassify fresh stillbirth as macerated stillbirth.[21]

Other factors that might have contributed to the reduction in CFRs were partograph use, improved fetal heart monitoring and having a dedicated maternity theatre to reduce waiting times for caesarean section.

The CFR for prematurity was reduced by more than half. One of the major interventions was the introduction of KMC, which has been shown to reduce mortality and morbidity in low-resource settings by 40% and 34%, respectively.[13] Moreover, CPAP was also introduced, which has been reported to improve survival of preterm babies by 27% among resource-limited settings.[22] Finally, in 2014 surfactant administration was introduced, which may have further reduced the mortality in prematurity-related deaths. Although surfactant is not yet provided in many developing countries as demonstrated in this study, it can be given safely under the supervision of experienced personnel.

This study suggests that a significant reduction in CFR for intrapartum-related hypoxia can be expected to occur after the implementation of multiple interventions to improve the inpatient care of small and sick newborns. It further confirms the findings of a Delphi study that estimated supportive care in a special care baby unit could avert 70% of neonatal deaths due to preterm birth complications, and that 90% could be averted with the availability of neonatal intensive care units.[23]

The fact that only 2.3% of early neonatal deaths were attributed to infection is a surprising finding. In another study, infections accounted for around 14% of early neonatal deaths.[24] Infection is the most common in the lowest socioeconomic groups and this could be because most cases of neonatal sepsis were late neonatal deaths (after 1 week) and so would be missed by the audit. Furthermore, 15.4%

of early neonatal deaths had an unknown cause of death. It is possible that this represents an underestimation of the proportion of deaths attributable to infection. In addition, some cases of respiratory distress syndrome may actually have been cases of neonatal infection, as it is often very difficult to differentiate between the two conditions without sophisticated diagnostic tests.

The relatively large sample size in this study provided a good opportunity to document the distribution of the causes and factors contributing to perinatal mortality. Nevertheless, in 43% of the deaths, a cause of death could not be established. This proportion was highest for macerated stillbirths (88% unknown) with 49% of fresh stillbirths and 15% of early neonatal deaths assigned to the category 'unknown'. This may be related to the availability of information about the deaths: the longer the baby died before birth, the more difficult it becomes to obtain information about the circumstances of the death. Furthermore, diagnosis is easier for newborns than for stillbirths since clinicians usually have more time to examine and monitor a sick newborn baby during the period of admission and before death. Another possible reason for the inability to establish the cause of death may be the lack of diagnostic capacity, relating to both staff and equipment. To reduce the proportion of deaths with an unknown cause in low-resource settings, healthcare providers should keep clinical records as complete and as accurate as possible.

### Effectiveness of perinatal death audit

Ideally, the results of this study should be confirmed by randomised controlled trials (RCT) conducted in the same context of care.

Other reports found that perinatal death audit did not improve perinatal outcomes. In a 5-year study of health facilities in South Africa, it was observed that among the 54 health facilities that started perinatal death audit, only 30% had a decrease in perinatal mortality, while 35% had an increase.[25] The study identified several facility-related factors that contributed to the increase in perinatal mortality, including lack of use of antenatal steroids, lack of nursing personnel, fetal distress not detected during the antepartum period and incorrect use of the partograph. Additional ITS studies with longer preintervention time points of data collection in low-income and middle-income countries should confirm or refute our findings understanding of the effects of this intervention in these settings. Finally, RCTs using a stepped wedge design, are needed to assess the effect of perinatal death audit on perinatal outcomes.

A systematic review and meta-analysis of the effect of perinatal audits in low-resource settings showed a reduction of perinatal deaths of 30% attributable to this intervention.[6] A large RCT conducted in Mali and Senegal which aimed to assess the effect of a complex intervention (including maternal death audits) on maternal and perinatal outcomes, showed that maternal audits reduced ENMR (early neonatal mortality rate) by 24%, while there was no effect on SBR (stillbirth rate).[26] These findings on perinatal

outcomes are in line with those of our study, even though the intervention was different (we assessed perinatal death audits, while the RCT assessed maternal death audits). This suggests that although perinatal death audits seem promising in reducing perinatal mortality, the role of these audits on SBR and ENMR remains unclear.

## CONCLUSION

This study highlights the potential of perinatal death review to identify how care pathways can be improved over time to reduce adverse perinatal outcomes in low-resource settings. This study shows that the introduction of perinatal death audit allowed for the identification of the main causes of death and contributing factors, and prioritise areas in the provision of healthcare that need improvement. However, although the intervention showed a decrease in early neonatal death, the perinatal audit process did not have an impact on PMR (perinatal mortality rate) and SBR. These findings should encourage more healthcare providers to undertake research on perinatal death reviews, especially in low-resource settings to assess the effectiveness of perinatal death audit on perinatal deaths in general, but also on stillbirths and neonatal deaths in particular.

**Acknowledgements** We would like to thank all healthcare providers who took part in the review process. We particularly appreciate the contributions of Musana Othniel, Sebunya Robert, Nazziwa Rita, Catherine Nyagabyaki, Nabwami Immaculate, Tumwine Gilbert and Zaake. We are also grateful to Robert Lloyd for his contribution to data processing, to Sarah White for help with part of the statistical analysis, to Mathews Mathai for reviewing the manuscript and to Caroline Hercod for editing the manuscript.

**Contributors** VNK contributed to study design, data collection, analysis and writing up. MA processed and analysed the data and contributed to all drafts of the paper. JED contributed to the conception of the paper, to the analysis and writing of the paper. RB took part in writing the study proposal, data collection and review of cases. PO contributed to the study proposal, data collection and case reviews. NvdB oversaw the conception and analysis and contributed to the initial and final drafts of the paper.

**Funding** The cost of data processing, analysis and publication was sponsored by the 'Making it Happen' Programme funded by DFID/UKAid.

**Competing interests** None declared.

**Patient consent for publication** Not required.

**Ethics approval** Approval for the study was obtained from the Nsambya Hospital Ethical Review Committee.

**Provenance and peer review** Not commissioned; externally peer reviewed.

**Data availability statement** Data are available on reasonable request. The datasets used and/or analysed during the current study are available from the corresponding author on reasonable request.

**ORCID iD**
Mamuda Aminu http://orcid.org/0000-0002-2335-7147

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
