## [Reviewer comments · BMJ Open]

ARTICLE DETAILS

TITLE (PROVISIONAL)	Prospective Study to Explore Changes in Quality of Care and Perinatal Outcomes After Implementation of Perinatal Death Audit in Uganda
AUTHORS	Kirabira, Victoria; Aminu, Mamuda; Dewez, Juan; Byaruhanga, Romano; Okong, Pius; van den Broek, Nynke

VERSION 1 - REVIEW

REVIEWER	Sarah Saleem Aga Khan University, Karachi, Pakistan
REVIEW RETURNED	17-Dec-2018

GENERAL COMMENTS	Some of the comments are as follows; Abstract: under the heading of Results in second paragraph, last lines need further clarity starting as "Cause of death was unknown in 43.2% of cases. The proportion of deaths with unknown cause decreased from 34.1% (overall) in 2008 to 2.7% in 2013. the message is not clear and is confusing Instead of writing "data was" collected please write 'data were' collected and make this correction wherever needed Methods: under the heading of 'The Audit process' in the last lines of the last paragraph, it is suggested to mention few interventions as examples for readers' interest Under the results section, second paragraph, 4th line from above, starting as--' Between 23.2% and 39.4% were considered to have been born prematurely' this line is confusing and needs further clarity Under the heading of 'cause of death' the line starting as --- 'However, the proportion of deaths for which cause of death could not be established ("cause unknown") decreased from 34.1% (overall) in 2008 to 2.7% in 2013---' is not clear and confusing please address this. Please correct some typo mistakes "senior more as senior most". Main findings are repeated at the beginning of the discussion section, please check if it is the requirement of the journal, otherwise would suggest to avoid repeating the results here. Under the heading of strengths and limitations, it is suggested that authors also discuss overlapping confidence intervals of the rates at the beginning and after the implementation of the audit, also to comment on auditing one fourth of all the deaths and how this could have impacted the results especially for infection related deaths having very small numbers. For increase in the early neonatal deaths, there is a possibility that those babies who were resuscitated at the time of birth could not
--

	survey long and hence an increase in the early neonatal mortality rates, a comment on this by the authors would be useful
--	---

REVIEWER	Dr Merlin Willcox Department of Primary Care and Population Sciences, University of Southampton, UK I have collaborated with Dr Victoria Nakibuuka Kirabira and Dr Pius Okong in preparing a grant proposal.
REVIEW RETURNED	11-Jan-2019

GENERAL COMMENTS	Review of Prospective Study to Explore Changes in Quality of Care and Perinatal Outcomes After Implementation of Perinatal Death Audit in Uganda General comments Thank you for giving me the opportunity to review this very interesting manuscript about some very important work. This deserves to be published, but I would like to recommend a number of improvements to the manuscript. These are given in detail below. The major suggestions are: (1) If possible, I would like to recommend re-analysis using an interrupted time-series method. This would put this study into a higher quality category and would enable its inclusion in Cochrane reviews etc. (2) Much more reflection and self-critique in the limitations section about possible biases and other possible explanations for trends observed. (3) More explanation on rating of “quality of care” in order to explain why such low levels of suboptimal care were reported –and whether this changed over time. Detailed specific comments P3 Strengths and limitations: “This study has demonstrated how the introduction of perinatal death audit helped to identify the main causes of death and contributing factors and priorities action.” Is this supposed to be “priority actions” or “to prioritise actions”?- or you could delete the last bit since the next point talks about “priority areas”. Background para 3: “Similarly, in a systematic review focusing on low- and middle-income countries, a meta-analysis of 7 before-and-after studies reported up to 30% (95% confidence interval, 21%–38%) reduction in perinatal mortality after introduction of perinatal death audit in healthcare facilities [6].” This is not “up to 30%”. 30% is actually the average reduction, from meta-analysis of all the studies. The authors state “These studies were combined in a random effects metaanalysis with a resultant relative risk of 0.70 (95% CI, 0.62–0.79).” “low-resources settings” should be “low-resource settings” The research question(s) could be more clearly defined at the end of the Background. Last paragraph of Background and first paragraph of Methods: Description of study setting: This was not introduction in the whole of Kampala – it was in one
--

private hospital.

The setting should specify that this is a private not-for-profit hospital. It is not "THE referral centre for Kampala and Makindye Health Sub-districts." It is one of many referral hospitals. Makindye is a district of Kampala (the text above implies it is separate). It would be more accurate to state that Nsambya is "a private referral hospital in Makindye subdistrict, within Kampala". Arguably the main (government) referral centre in Kampala is Mulago Hospital. It is important for readers to understand this because what is possible to implement in a private setting may be more challenging to implement in an even busier and less well-resourced government setting, serving an even larger and poorer population.

What is meant by "high care area beds"? Is this a high dependency unit, or a neonatal intensive care?

Is surfactant administered to all premature babies who need it? My understanding was that surfactant is very expensive in Uganda and most patients (even in private hospitals) cannot afford it. Table 2 implies it was only introduced in Feb 2014. Be careful about making generalisations in the description of the facility, which do not apply to all patients and throughout the study period. In fact according to table 2, most of the interventions described in the "study setting" were NOT present before, and were only introduced as a result of the perinatal death audit, part way through this study. So they should probably be omitted from this description of the study setting.

Explain why only one in 4 deaths was audited. Don't national guidelines recommend that all deaths are audited? And why was random selection used, rather than purposive selection? For example, it could be argued that there is little to be gained from reviewing macerated stillbirths (as evidenced by the fact that even in this study, 88% of macerated stillbirths were from an unknown cause), and much more to be gained from auditing fresh stillbirths and neonatal deaths (which have much lower rates of unknown cause).

The audit process: "The quality of care provided was also graded by consensus and classified into four categories"

- Please clarify – presumably you were only rating quality of care at Nsambya hospital? So this does not include quality of care at previous facilities (primary or secondary) from which patients were referred?

- However table 2 row 1 suggests that delays in referral were considered.

Results

para 1: "Of these, 603 (23%) perinatal deaths were selected for audit. After review, 526 were found to meet the criteria for perinatal mortality." The methods section should make it clear what these criteria were. So what of the other 603-526 = 97 deaths? Why were they excluded?

This casts doubt on the overall figure of mortality, because effectively 16% (97/603) of "perinatal deaths" were deemed after review not to be perinatal deaths... This needs more explanation.

Trend in stillbirth rate – is it possible to separate this into trends for fresh and macerated stillbirths? Please provide this data.

Case fatality rates: is the increase in case fatality from infection statistically significant? The numbers are very small so it seems

improbable that this is statistically significant. If it is not statistically significant, it should not be called an “increase”.

The text in the figures is so small as to be almost illegible unless greatly magnified. Please improve the formatting and text size in all the figures.

Quality improvement line 1 (also first section of discussion): “adjudged” is a legal term and makes this sound like a trial. How about “deemed” –sounds less legalistic.

The rate of suboptimal care is surprisingly low. This implies that you were only looking at quality of care in Nsambya hospital, not overall, including facilities from which patients had been referred. However table 2 row 1 suggests that referral delays were considered. Please clarify whether this is included in the % of “suboptimal care”.

Was there any trend in % of cases with suboptimal care over time? It would be very useful to analyse this. If the hypothesis is correct that death audit is effective, and if deaths were selected truly at random, the % of deaths related to suboptimal care would be expected to decrease over time.

“In 2007 and 2008, the perinatal death review team observed many cases of delay between the decision for, and actual conduct of, Caesarean Section which was associated with high numbers intrapartum-related hypoxia” – above, it is stated that the death reviews started in 2008. Did they start in 2007 or 2008? Should be “high numbers OF intrapartum-related hypoxia”

The text says “rescue surfactant was made available for use in 2010.” However table 2 and the discussion say that this was implemented in 2014. Please clarify. Also was this made available to all babies who needed it, or only those whose parents could afford to pay for it? If the latter was the case (as I suspect), what % of babies who needed it actually received it? If the former, how was it funded?

“No additional specific actions to improve the management of neonatal infections were documented as this was not considered a major cause of mortality”. – this calls into question the death review process. Does this mean that no cases of neonatal infections were audited / reviewed? Or that all of those audited / reviewed were deemed to have received optimal care? Did all babies who died of sepsis receive treatment according to the guidelines? Were the correct antibiotics always available and given in time? Were all the babies monitored optimally and treated at the first signs of sepsis without delay? As case fatality increased from infections over the study period, there could have been some problems.

Whether something is a “major” cause of mortality is not a consideration when reviewing individual cases. Each case should be discussed on its merits, and recommendations made accordingly. Whether something is a “major” cause of mortality is not (usually) part of the discussion. The fact that few cases of sepsis were detected is also unusual. Globally, and particularly in low-income countries, sepsis is one of the main causes of neonatal deaths. Is it possible that some cases of sepsis were missed, or misdiagnosed, or hidden among the “unknown” causes?

Discussion

Strengths and limitations sentence 1: what is meant by “priorities action” (see comment above also)? Should this be “priority actions” or “ priorities for action”?

“There are few studies exploring the use of perinatal death audit to improve care in low-resource settings” – there are actually quite a few, for example the Pattinson (2009) review to which the authors refer in the introduction found 10 studies, of which 7 could be meta-analysed. However, all of these were deemed to be of low quality as they are uncontrolled before-and-after studies. Unfortunately the present study would also fall into the same category, UNLESS the authors are able to produce an interrupted time-series analysis. The Cochrane EPOC group (Effective Practice and Organisation of Care) will allow inclusion of randomised controlled trials, controlled before-and-after studies, and interrupted time series in its reviews. All other designs are deemed to be of low quality and cannot be included in Cochrane reviews. Therefore when Pattinson did a Cochrane review on the same topic, he actually found NO studies which met the Cochrane inclusion criteria. So the problem is not about quantity of studies, it is about methodological quality.

Interrupted time series must have at least 3 measurements before and 3 measurements after introduction of the intervention, which must start at a clearly defined point in time. It is possible that the authors may be able to re-analyse their results as an interrupted time-series, particularly if they are able to provide measurements of perinatal mortality every 6 months (which would then fit the Cochrane criteria of at least 3 measurements before introduction of the intervention). I would strongly encourage the authors to do this if at all possible, because it would mean that their study could be included in future Cochrane reviews, and would be considered of higher quality.

If the authors are not able to do this (or even if they are), there is a lot more to be written about limitations. For example consider the following Cochrane EPOC group criteria for assessing risk of bias in interrupted time series studies:

- was the intervention independent of other changes?
- was the shape of the intervention effect prespecified?
- was the intervention unlikely to affect data collection?
- was knowledge of the allocated interventions adequately prevented during the study?
- were incomplete outcome data adequately addressed?
- was the study free from selective outcome reporting?
- was the study free from other risks of bias?

“Problems identified through audit”, “interventions for prematurity-related complications”, – these sections of the discussion are largely a repetition of the results (and relevant results would be better in the results section if not presented before). No need to repeat the same thing. I would suggest changing the title to “comparison with published literature” and focus on comparison with other studies.

Surfactant – see comments above – did it start in 2010 or 2014?
Was it available for all?

Section on effectiveness of perinatal death audit – why focus on the study in the Netherlands? There has only been one cluster-randomised trial of the effectiveness of perinatal death audit,

	conducted in France: Dupont C, Winer N, Rabilloud M, Touzet S, Branger B, Lansac J, Gaucher L, Duclos A, Huissoud C, Boutitie F, Rudigoz R, Colin C, and the Opera group.. Multifaceted intervention to improve obstetric practices: The OPERA cluster-randomized controlled trial. European Journal of Obstetrics, Gynecology, & Reproductive Biology 2017;215:206-212. [DOI: https://doi.org/10.1016/j.ejogrb.2017.06.026] The authors should quote this as a priority. It also showed a reduction in suboptimal care but not in mortality. However it is much harder to show a change in mortality in a high-income setting where mortality is already very low. “Further research” – I would suggest it is not just the proportion which needs to be reviewed, but how the cases should be selected. For example, in hindsight, do the authors think that their method of selection (of every 4th case) was optimal? Or could there be better methods of selecting cases which give a higher yield of useful recommendations? I would also suggest that there is a question about cost-effectiveness. Can the authors provide any information on how much it cost to conduct the audit? (in terms of staff time, cost of implementing recommendations, etc)? Supplementary table 1: please provide absolute numbers as well as rates. Were the rates calculated using overall data on deaths as opposed to those selected for audit / review? Supplementary table 4: as mentioned above, please could you split this into time period – at least before and after introduction of the audit, and if possible into more time periods?
--	---

REVIEWER	Maryam Mozooni The University of Western Australia Australia
REVIEW RETURNED	16-Feb-2019

GENERAL COMMENTS	Thank you for the opportunity to review this paper. Kirabira and colleagues, in this study, described the changes observed in perinatal outcomes after introduction of perinatal death audit in a tertiary hospital in Kampala and explored the trends in case fatality rates of the major causes of perinatal mortality identified. Although auditing perinatal mortality is an important topic, meaningful conclusions expected from a research paper cannot be drawn from the methodology of this study and the presented analyses. Data management and analysis are, in fact, poorly explained. Extensive revisions in Abstract and other sections of the manuscript and undertaking comparative analysis are required for drawing inference from these data. Proofreading and correction of occasional typo/ English errors is also suggested. No reference for ethics approval was provided although it was stated that "Approval for the study was obtained from the Nsambya Hospital Ethical Review Committee." Comments: Major:  1. Title of the paper is not correct. From what has been described in Methods section of the study, it is understood that the study is a 'retrospective' and not a prospective study: P5-Line 52: Authors explained that "The data was obtained and aggregated monthly through a retrospective analysis of registers and patient records." 2. It is unclear if the participants of the study were limited to
--

	"Perinatal deaths from 2006 to 2015", as currently has been described. If that is correct, how the perinatal mortality rates have been calculated? Did the authors have access to all births (the denominator), which enabled them to calculate these rates? If yes, the Methods section of the paper should be corrected in relevant sections. 3. The results presented here are only from descriptive analysis and no analytical measures for comparison of rates have been undertaken or shown. Statistical analysis results for comparison of the rates with each other should be provided. Presenting 95% CI or P-value of that comparison is required to show whether such difference in rates are statistically meaningful at all or there is no significant difference between them (before and after the audit). The process should then be described in the Methods section of the manuscript under a heading 'Statistical methods'. 4. The conclusions written are not justified by this study. To draw meaningful conclusions, more detailed data and comparative statistical analyses are required. 5. Strength and limitations of the study are poorly described.
--	---

REVIEWER	Lennarth Nyström Department of Epidemiology and Global Health Umeå University Umeå Sweden
REVIEW RETURNED	16-Mar-2019

GENERAL COMMENTS	This is an important and relevant study, however, the statistical analysis and the presentation of the results require revision Major concern The analysis of the "Trend in perinatal mortality" is confusing. Several statements such as "downward trend", "ENMR showed a slight increase", "PNMR showed a slowly? decreasing trend" etc are not supported by the data presented as point estimate and 95% CIs. In Figure 1 and 3 the plot of PNMR etc 2006-2015 is supplemented by a line (regression line?), but there is no information in the Statistical method section on regression analysis? There is no discussion on power of the study. Minor comments Age and parity of mothers should be presented as median and range (Median parity is given, but combined with the SD?). Precision of estimates: Limit the estimates to 2 digits e.g. Stillbirth rate 32.9 (95% CI:25.0-41.5) should be 33 (95% CI: 25-42), p=0.872 should be p=0.87 etc Figure 1. Regression (?) lines should be thicker, font size of scale on the x, y and z axis larger to be visible Figure 2. The 3rd pie graph contains too many wedges (max 6), increase size of text! Figure 3. See comments on Figure 1. Add variable names on x and y axis Supplementary table 2: Fresh stillbirths=125, but only 124 in the column! In general the legend to the figures and head of tables should contain information so the figure/table can be understandable without reading the whole article.
--

VERSION 1 – AUTHOR RESPONSE

Reviewer: 1

COMMENT	RESPONSE
Abstract: under the heading of Results in second paragraph, last lines need further clarity starting as "Cause of death was unknown in 43.2% of cases. The proportion of deaths with unknown cause decreased from 34.1% (overall) in 2008 to 2.7% in 2013. the message is not clear and is confusing	Edited to improve clarity. Thank you.
Instead of writing "data was" collected please write 'data were' collected and make this correction wherever needed	Thank you for this observation. However, "data" can be used either in singular or in plural sense. Here, it was used here as a singular mass noun.
Methods: under the heading of 'The Audit process' in the last lines of the last paragraph, it is suggested to mention few interventions as examples for readers' interest	Example provided. See the referred section (font in red).
Under the results section, second paragraph, 4th line from above, starting as--' Between 23.2% and 39.4% were considered to have been born prematurely' this line is confusing and needs further clarity	Edited. Thank you.
Under the heading of 'cause of death' the line starting as - --'However, the proportion of deaths for which cause of death could not be established ("cause unknown") decreased from 34.1% (overall) in 2008 to 2.7% in 2013--- ' is not clear and confusing please address this.	Edited. Thank you.
Please correct some typo mistakes "senior more as senior most'.	Corrected.
Main findings are repeated at the beginning of the discussion section, please check if it is the requirement of the journal, otherwise would suggest to avoid repeating the results here.	Thank you. This format is in line with the STROBE reporting guidelines, which the journal uses.
Under the heading of strengths and limitations, it is suggested that authors also discuss overlapping confidence intervals of the rates at the beginning and after the implementation of the audit, also to comment on auditing one fourth of all the deaths and how this could have impacted the results especially for infection related deaths having very small numbers.	The CI is now discussed (see "Strengths and limitations"). Since it was a systematic random sampling, we believe that auditing 25% of the cases would have minimal (if any) effect on the overall distribution of the cause of death.
For increase in the early neonatal deaths, there is a possibility that those babies who were resuscitated at the time of birth could not survive long and hence an increase in the early neonatal mortality rates, a comment on this by the authors would be useful.	This is unclear. Since resuscitation can only be used to save live babies, we cannot see how its use could increase early neonatal death. Stillborn babies cannot be resuscitated.

Reviewer: 2

COMMENT	RESPONSE
General comments	

COMMENT	RESPONSE
Thank you for giving me the opportunity to review this very interesting manuscript about some very important work. This deserves to be published, but I would like to recommend a number of improvements to the manuscript. These are given in detail below.	Thank you.
The major suggestions are:	
(1) If possible, I would like to recommend re-analysis using an interrupted time-series method. This would put this study into a higher quality category and would enable its inclusion in Cochrane reviews etc.	Thank you for this suggestion. ITS done and now included.
(2) Much more reflection and self-critique in the limitations section about possible biases and other possible explanations for trends observed.	Reviewed and updated. Thank you.
(3) More explanation on rating of “quality of care” in order to explain why such low levels of suboptimal care were reported –and whether this changed over time.	Grading method provided (See “audit process” under Methods). Thank you.
Detailed specific comments:	
P3 Strengths and limitations: “This study has demonstrated how the introduction of perinatal death audit helped to identify the main causes of death and contributing factors and priorities action.” Is this supposed to be “priority actions” or “to prioritise actions”?- or you could delete the last bit since the next point talks about “priority areas”.	Edited. Thank you.
Background para 3: “Similarly, in a systematic review focusing on low- and middle-income countries, a meta-analysis of 7 before-and-after studies reported up to 30% (95% confidence interval, 21%–38%) reduction in perinatal mortality after introduction of perinatal death audit in healthcare facilities [6].” This is not “up to 30%”. 30% is actually the average reduction, from meta-analysis of all the studies. The authors state “These studies were combined in a random effects meta-analysis with a resultant relative risk of 0.70 (95% CI, 0.62–0.79)”.	Edited. Thank you.
“low-resources settings” should be “ low-resource settings”	Edited. Thank you.
The research question(s) could be more clearly defined at the end of the Background.	The last paragraph in the background states the study objective, i.e. to “describe changes in perinatal outcomes observed after the introduction of perinatal death audit and explore the trends in case fatality rates for the major identified causes of perinatal mortality”
Description of study setting:	
This was not introduction in the whole of Kampala – it was in one private hospital.	Edited. Thank you.
The setting should specify that this is a private not-for-profit hospital. It is not “THE referral centre for Kampala and Makindye Health Sub-districts.” It is one of many referral hospitals. Makindye is a district	Edited to reflect this comment. Thank you.

COMMENT	RESPONSE
of Kampala (the text above implies it is separate). It would be more accurate to state that Nsambya is “a private referral hospital in Makindye subdistrict, within Kampala”. Arguably the main (government) referral centre in Kampala is Mulago Hospital. It is important for readers to understand this because what is possible to implement in a private setting may be more challenging to implement in an even busier and less well-resourced government setting, serving an even larger and poorer population.	
What is meant by “high care area beds”? Is this a high dependency unit, or a neonatal intensive care?	It is a newborn unit that has high dependency beds and 10 neonatal intensive care beds – changes made to the section. Thank you.
Is surfactant administered to all premature babies who need it? My understanding was that surfactant is very expensive in Uganda and most patients (even in private hospitals) cannot afford it. Table 2 implies it was only introduced in Feb 2014. Be careful about making generalisations in the description of the facility, which do not apply to all patients and throughout the study period. In fact according to table 2, most of the interventions described in the “study setting” were NOT present before, and were only introduced as a result of the perinatal death audit, part way through this study. So they should probably be omitted from this description of the study setting.	Surfactant has been available at the facility since 2014. Not all preterm babies get it. We use the rescue method where only those that need it are given according the recent guidelines for management of RDS - Changes made page 5
Explain why only one in 4 deaths was audited. Don’t national guidelines recommend that all deaths are audited? And why was random selection used, rather than purposive selection? For example, it could be argued that there is little to be gained from reviewing macerated stillbirths (as evidenced by the fact that even in this study, 88% of macerated stillbirths were from an unknown cause), and much more to be gained from auditing fresh stillbirths and neonatal deaths (which have much lower rates of unknown cause).	We recognize the importance of auditing all deaths. However, due to high numbers of death, this was not practically possible. The WHO Perinatal Death Audit Guidelines provide for such situations. It recommends sampling the cases, with more focus on neonatal deaths and “fresh” stillbirths. In this study, half of the cases were neonatal deaths and there were more fresh than macerated stillbirths. Nonetheless, we could have included more fresh stillbirths. Systematic random sampling was used to reduce the chances of selection bias. Moreover, we needed to find the top causes of death to study their CFR. A purposive sampling may not have given a fair picture of the distribution of cause of death to enable identification of the top causes.
The audit process: “The quality of care provided was also graded by consensus and classified into four categories” - Please clarify – presumably you were only rating quality of care at Nsambya hospital? So this does not include quality of care at previous facilities (primary or secondary) from which patients were referred?	Thank you for this observation. We have edited the section for better clarity.

COMMENT	RESPONSE
- However, table 2 row 1 suggests that delays in referral were considered.	
Results	
para 1: "Of these, 603 (23%) perinatal deaths were selected for audit. After review, 526 were found to meet the criteria for perinatal mortality." The methods section should make it clear what these criteria were. So what of the other 603-526 = 97 deaths? Why were they excluded?	What healthcare providers document as perinatal mortality does not always meet the technical definition of the term. While they documented 603 as perinatal deaths, a closer look showed that only 526 actually met the definition. Hence, the removal of the remaining 97 cases.
This casts doubt on the overall figure of mortality, because effectively 16% (97/603) of "perinatal deaths" were deemed after review not to be perinatal deaths... This needs more explanation.	This could not affect perinatal mortality rate, because in calculation of the perinatal mortality rate we included all stillbirths, early neonatal deaths and the deliveries. We did not use the perinatal deaths audited to calculate the overall perinatal mortality rate
Trend in stillbirth rate – is it possible to separate this into trends for fresh and macerated stillbirths?	The trend for both types of stillbirth is similar, with each type of stillbirth contributing similar numbers (FSB=1062, MSB=1087, from 2006 to 2015). Thus, we feel this will be redundant. However, we've provided data for each type separately where the distinction gives additional information. (See Supplementary Tables)
Case fatality rates: is the increase in case fatality from infection statistically significant? The numbers are very small, so it seems improbable that this is statistically significant. If it is not statistically significant, it should not be called an "increase".	The increase infection CFR was not significant. p-value provided (p=0.1).
The text in the figures is so small as to be almost illegible unless greatly magnified. Please improve the formatting and text size in all the figures.	Edited. Thank you.
Quality improvement line 1 (also first section of discussion): "adjudged" is a legal term and makes this sound like a trial. How about "deemed" –sounds less legalistic.	Edited.
The rate of suboptimal care is surprisingly low. This implies that you were only looking at quality of care in Nsambya hospital, not overall, including facilities from which patients had been referred. However table 2 row 1 suggests that referral delays were considered. Please clarify whether this is included in the % of "suboptimal care".	We only included sub optimal care at Nsambya, not the referring units. If there was a delay in referral from the referring unit then it was considered as an avoidable factor and means to communicate to the lower level health unit would be done
Was there any trend in % of cases with suboptimal care over time? It would be very useful to analyse this. If the hypothesis is correct that death audit is effective, and if deaths were selected truly at random, the % of deaths related to suboptimal care would be expected to decrease over time.	Cases were selected using systematic sampling (see previous response). We did not analyse trends of suboptimal as this was based on healthcare providers' perspective and would not be a reliable outcome indicator of quality. We believe that perinatal deaths and CFRs are much more appropriate and reliable in this study.
"In 2007 and 2008, the perinatal death review team observed many cases of delay between the decision for, and actual conduct of, Caesarean Section which	The audit process was started in 2008, though data collection on the outcome indicators started earlier (2006).

COMMENT	RESPONSE
was associated with high numbers intrapartum-related hypoxia” – above, it is stated that the death reviews started in 2008. Did they start in 2007 or 2008?	
Should be “high numbers OF intrapartum-related hypoxia”	Edited. Thank you.
The text says, “rescue surfactant was made available for use in 2010.” However, table 2 and the discussion say that this was implemented in 2014. Please clarify. Also, was this made available to all babies who needed it, or only those whose parents could afford to pay for it? If the latter was the case (as I suspect), what % of babies who needed it actually received it? If the former, how was it funded?	Surfactant administration was started in 2014, Surfactant is a product that is paid for in our hospital just like other commodities like antibiotics or caesarean section. The cost of surfactant is less than an operation like caesarean section. Majority (98%) of the preterms that needed it were able to pay for it and actually received it This is a private not for profit hospital, majority of the patients that are admitted are able to afford the rates offered by the hospital. We also get low cost surfactant from south Africa / India which costs about 200 dollars per vial.
“No additional specific actions to improve the management of neonatal infections were documented as this was not considered a major cause of mortality”. – this calls into question the death review process. Does this mean that no cases of neonatal infections were audited / reviewed? Or that all of those audited / reviewed were deemed to have received optimal care? Did all babies who died of sepsis receive treatment according to the guidelines? Were the correct antibiotics always available and given in time? Were all the babies monitored optimally and treated at the first signs of sepsis without delay? As case fatality increased from infections over the study period, there could have been some problems.	The number of cases for sepsis is low, this could be because majority of the sepsis usually occur as late on set. Early onset sepsis may present with other co morbidities like Respiratory distress syndrome, seizure with asphyxia. Perinatal death audit usually is done for deaths occurring within the first week and therefore causes of death due to sepsis may be low for perinatal deaths. Furthermore, the diagnosis of sepsis is challenging in our setting, Blood culture is the gold standard for diagnosis of sepsis, but this is not routinely done. Hence this may make the numbers low.
Whether something is a “major” cause of mortality is not a consideration when reviewing individual cases. Each case should be discussed on its merits, and recommendations made accordingly. Whether something is a “major” cause of mortality is not (usually) part of the discussion. The fact that few cases of sepsis were detected is also unusual. Globally, and particularly in low-income countries, sepsis is one of the main causes of neonatal deaths. Is it possible that some cases of sepsis were missed, or misdiagnosed, or hidden among the “unknown” causes?	I agree the numbers of sepsis are low, because we do not do routine blood cultures this may reduce to the number of sepsis diagnosed. Perinatal death audits are done in the first one week of life, and yet a baby with die after the first one week, hence this may be missed .
Discussion	
Strengths and limitations sentence 1: what is meant by “priorities action” (see comment above also)? Should this be “priority actions” or “ priorities for action”?	Edited. Thank you.
“There are few studies exploring the use of perinatal	Thank you for this suggestion. ITS done and now

COMMENT	RESPONSE
death audit to improve care in low-resource settings” – there are actually quite a few, for example the Pattinson (2009) review to which the authors refer in the introduction found 10 studies, of which 7 could be meta-analysed. However, all of these were deemed to be of low quality as they are uncontrolled before-and-after studies. Unfortunately the present study would also fall into the same category, UNLESS the authors are able to produce an interrupted time-series analysis. The Cochrane EPOC group (Effective Practice and Organisation of Care) will allow inclusion of randomised controlled trials, controlled before-and-after studies, and interrupted time series in its reviews. All other designs are deemed to be of low quality and cannot be included in Cochrane reviews. Therefore when Pattinson did a Cochrane review on the same topic, he actually found NO studies which met the Cochrane inclusion criteria. So the problem is not about quantity of studies, it is about methodological quality. Interrupted time series must have at least 3 measurements before and 3 measurements after introduction of the intervention, which must start at a clearly defined point in time. It is possible that the authors may be able to re-analyse their results as an interrupted time-series, particularly if they are able to provide measurements of perinatal mortality every 6 months (which would then fit the Cochrane criteria of at least 3 measurements before introduction of the intervention). I would strongly encourage the authors to do this if at all possible, because it would mean that their study could be included in future Cochrane reviews, and would be considered of higher quality.	included.
If the authors are not able to do this (or even if they are), there is a lot more to be written about limitations. For example, consider the following Cochrane EPOC group criteria for assessing risk of bias in interrupted time series studies:  • was the intervention independent of other changes? • was the shape of the intervention effect prespecified? was the intervention unlikely to affect data collection? was knowledge of the allocated interventions adequately prevented during the study? were incomplete outcome data adequately addressed? was the study free from selective outcome reporting? was the study free from other risks of bias?	Thank you for this suggestion. ITS done and now included.

COMMENT	RESPONSE
“Problems identified through audit”, “interventions for prematurity-related complications”, – these sections of the discussion are largely a repetition of the results (and relevant results would be better in the results section if not presented before). No need to repeat the same thing. I would suggest changing the title to “comparison with published literature” and focus on comparison with other studies.	Thank you. The two sections are talking about different things and relate to the existing literature. Thus, it wouldn't be appropriate to be moved to the results section.
Surfactant – see comments above – did it start in 2010 or 2014? Was it available for all?	Corrected (see previous response). Thank you.
Section on effectiveness of perinatal death audit – why focus on the study in the Netherlands? There has only been one cluster-randomised trial of the effectiveness of perinatal death audit, conducted in France: Dupont C, Winer N, Rabilloud M, Touzet S, Branger B, Lansac J, Gaucher L, Duclos A, Huissoud C, Boutitie F, Rudigoz R, Colin C, and the Opera group.. Multifaceted intervention to improve obstetric practices: The OPERA cluster-randomized controlled trial. European Journal of Obstetrics, Gynecology, & Reproductive Biology 2017;215:206-212. [DOI: https://doi.org/10.1016/j.ejogrb.2017.06.026] The authors should quote this as a priority. It also showed a reduction in suboptimal care but not in mortality. However, it is much harder to show a change in mortality in a high-income setting where mortality is already very low.	Reviewed with additional literature. Thank you.
“Further research” – I would suggest it is not just the proportion which needs to be reviewed, but how the cases should be selected. For example, in hindsight, do the authors think that their method of selection (of every 4th case) was optimal? Or could there be better methods of selecting cases which give a higher yield of useful recommendations? I would also suggest that there is a question about cost-effectiveness.	As discussed in a response to a previous comment, we believe that every case has a lesson to be learnt and should be reviewed. However, when this is impractical, the systematic sampling we used was helpful in reducing the chances of selection bias. Because of the large number of cases, we also believe the results obtained from the samples is a fair representation of the true picture of the quality of care.
Can the authors provide any information on how much it cost to conduct the audit? (in terms of staff time, cost of implementing recommendations, etc)?	Thank you. Although this was not part of our objectives, the audit was conducted by a multidisciplinary audit team. Team meetings were held weekly. However, we did not track the cost of the implementation.
Supplementary table 1: please provide absolute numbers as well as rates. Were the rates calculated using overall data on deaths as opposed to those selected for audit / review?	Numbers provided.
Supplementary table 4: as mentioned above, please could you split this into time period – at least before and after introduction of the audit, and if possible into more time periods ?	Supplementary Table 4 shows categories of standards of care. We did not analyse trends of standard of care using this categorisation as an indicator as it was based on healthcare providers' perspective and would not be a reliable outcome

COMMENT	RESPONSE
	indicator of quality. We believe that perinatal deaths and CFRs are much more appropriate and reliable in this study.

Reviewer: 3

COMMENT	RESPONSE
Thank you for the opportunity to review this paper. Kirabira and colleagues, in this study, described the changes observed in perinatal outcomes after introduction of perinatal death audit in a tertiary hospital in Kampala and explored the trends in case fatality rates of the major causes of perinatal mortality identified. Although auditing perinatal mortality is an important topic, meaningful conclusions expected from a research paper cannot be drawn from the methodology of this study and the presented analyses.	This is unclear. The paper aimed to report observed changes in perinatal outcomes before and after an intervention. This is exactly what the paper reports
Data management and analysis are, in fact, poorly explained.	The reviewer is unclear about what specific areas are "poorly explained". However, the methodology has been improved throughout.
Extensive revisions in Abstract and other sections of the manuscript and undertaking comparative analysis are required for drawing inference from these data.	No specific issues have been highlighted by the reviewer for improvement, but revisions have been done in all the said areas.
Proofreading and correction of occasional typo/ English errors is also suggested.	This has been done throughout the manuscript.
No reference for ethics approval was provided although it was stated that "Approval for the study was obtained from the Nsambya Hospital Ethical Review Committee."	This is correct.
Major Comments:	
1. Title of the paper is not correct. From what has been described in Methods section of the study, it is understood that the study is a 'retrospective' and not a prospective study: P5-Line 52: Authors explained that "The data was obtained and aggregated monthly through a retrospective analysis of registers and patient records."	Audit, by its nature is a retrospective process. However, its processes can be established before the deaths occur (prospectively) and this was what was done in this case.
2. It is unclear If the participants of the study were limited to "Perinatal deaths from 2006 to 2015", as currently has been described. If that is correct, how the perinatal mortality rates have been calculated? Did the authors have access to all births (the denominator), which enabled them to calculate these rates? If yes, the Methods section of the paper should be corrected in relevant sections.	To calculate PNM rates, you simply need the number of PNM and the total births. This data was available for the calculation. A line is added in the "Data collection" section of the methods mentioning this.
3. The results presented here are only from descriptive analysis and no analytical measures for comparison of rates have been undertaken or shown. Statistical analysis results for comparison of the rates with each	This has been done.

COMMENT	RESPONSE
other should be provided. Presenting 95% CI or P-value of that comparison is required to show whether such difference in rates are statistically meaningful at all or there is no significant difference between them (before and after the audit). The process should then be described in the Methods section of the manuscript under a heading 'Statistical methods'.	
4. The conclusions written are not justified by this study. To draw meaningful conclusions, more detailed data and comparative statistical analyses are required.	Reviewed.
5. Strength and limitations of the study are poorly described.	Improvement done

Reviewer: 4

COMMENT	RESPONSE
This is an important and relevant study, however, the statistical analysis and the presentation of the results require revision	Results re-analysed using interrupted time series (ITS).
Major concern	---
The analysis of the "Trend in perinatal mortality" is confusing. Several statements such as "downward trend", "ENMR showed a slight increase", "PNMR showed a slowly? decreasing trend" etc are not supported by the data presented as point estimate and 95% CIs. In Figure 1 and 3 the plot of PNMR etc 2006-2015 is supplemented by a line (regression line?), but there is no information in the Statistical method section on regression analysis?	Results re-analysed (ITS).
There is no discussion on power of the study.	Reviewed to reflect the new ITS results.
Minor comments	
Age and parity of mothers should be presented as median and range (Median parity is given, but combined with the SD?).	Reviewed.
Precision of estimates: Limit the estimates to 2 digits e.g. Stillbirth rate 32.9 (95% CI:25.0-41.5) should be 33 (95% CI: 25 -42), p=0.872 should be p=0.87 etc	Thanks for your comment. We believe the one decimal provided is more precise than the approximation.
Figure 1. Regression (?) lines should be thicker, font size of scale on the x, y and z axis larger to be visible	Comment no longer relevant due to re-analysis (ITS).
Figure 2. The 3rd pie graph contains too many wedges (max 6), increase size of text!	Reviewed. Thank you.
Figure 3. See comments on Figure 1. Add variable names on x and y axis	Comment no longer relevant due to re-analysis (ITS).
Supplementary table 2: Fresh stillbirths=125, but only 124 in the column!	Corrected. Thank you.
In general the legend to the figures and head of tables should contain information so the figure/table can be understandable without reading the whole article.	Tables and figures reviewed. Thank you.

VERSION 2 – REVIEW

REVIEWER	Merlin Willcox University of Southampton, UK I have written grant proposals with Drs Nakibuuka / Kirabira, Aminu and Okong.
REVIEW RETURNED	19-Jun-2019

GENERAL COMMENTS	General comments: The authors have now addressed the majority of the comments, and have performed an ITS analysis, for which they are to be congratulated. I am not an expert statistician so it may be worth asking a statistician to check they are happy with the analysis, but it looks fine to me. However the results do raise some additional questions which need to be addressed, particularly in the limitations section of the discussion, which needs to be expanded. Firstly, the article does not address all of the points on the Cochrane checklist for assessing quality of ITS studies: - was the intervention independent of other changes?- was the shape of the intervention effect prespecified?- was the intervention unlikely to affect data collection?- was knowledge of the allocated interventions adequately prevented during the study?- were incomplete outcome data adequately addressed?- was the study free from selective outcome reporting?- was the study free from other risks of bias? Secondly, the conclusions are somewhat surprising – that the audit had a massive impact on early neonatal death rates, but no effect on perinatal deaths overall or on stillbirths. The authors need to discuss whether they believe this finding, or whether it could be due to the fact that there were only 2 time points before the introduction of the intervention (and so some random variation could have accounted for the “before” changes – Cochrane specifies that there should be at least 3 timepoints before the intervention); or limited numbers. What further research would they recommend? Would they recommend that guidelines are changed so that audits focus only on early neonatal deaths, as audits of stillbirths appear to have had no impact? The discussion rightly mentions many of the interventions implemented – however a sceptic could read this and conclude that actually the audit was not needed because most of the interventions could have been predicted purely from cause of death (eg from the fact that hypoxia is the main cause, one could conclude that staff need training on use of partograms, and neonatal resus). The authors need to give more information on whether they think the audit itself added to the impact, or whether they could have achieved the same impact just by implementing these training interventions, without doing the audit. The article reads well and is almost ready for publication but there are a number of typos which need to be corrected (see below) and the font on the pie charts is still too small to be legible. Specific comments: Abstract: Conclusion – correct spelling of “introduction” Article summary, last line: “The intervention did not had an effect” – should be “The intervention did not have an effect”
--

P5 line 34 "numbers of death," should be "numbers of deaths," or "due to the large number of deaths"

P6 top of page – explain which types of professional were in the audit team. Was it just neonatologists / neonatal nurses and paediatricians, or did it also include obstetricians / midwives? Was the quality of care just regarding the paediatric quality of care, or was the quality of obstetric care / partogram use also considered?

P6 line 31 correct spelling of "Caesarean"

P6 last line "describe various modifications to the model which can be considered." – this sounds like an instruction or comment which has been left in by mistake? Please check.

P8 last line "Among the different types of mortalities, sub-optimal care was most common among early neonatal deaths the most (11.8%)" – delete "the most" at the end

P10 line 28 correct spelling of "intervention"

P10 line 49 "This study assessed then introduction of pernatal audits" – correct to "This study assessed the introduction of perinatal audits"

P10 line 54 correct spelling of "moreover"

Strengths and limitations section should mention the fact that a major limitation is that there were only 2 timepoints of data BEFORE introduction of the audits. Cochrane standards require there to be at least 3 timepoints before the intervention starts. The problem is that if there was some random variation this could lead to misinterpretation of the results. So: it is possible that actually there WAS an effect of the audits on overall perinatal mortality, but there was a random decrease between 2006 and 2007. Also, it is possible that the effect on early neonatal deaths may not have been as pronounced as suggested by the analysis, because the increase between 2006 and 2007 may have been in part due to random variation from year to year. So we should be cautious in interpreting the results.

Also – it is worth mentioning that when audits are first introduced, very often the mortality rate appears to increase in the first year or two, simply because of improved recording of deaths. This has been shown in numerous studies on perinatal death audits.

Another possible limitation is the lack of external input to the audit process. The methods section does not mention who was on the audit team, but presumably it was only staff from the hospital? It is to be expected that doctors, nurses and midwives often find it difficult to be self-critical and so may not have reported as many problems with quality of care as may have been identified by an external expert.

P11 line 22: "poor neonatal resuscitation skills, and, poor management of preterm babies"- remove commas – should be "poor neonatal resuscitation skills and poor management of preterm babies"

P11 lines 31-32: "The interventions for improvement included: training in the use of the partograph, fetal heart monitoring, neonatal resuscitation, measures to reduce delay in performing a cesarean section by involving doctors in the timely preparation of mothers before transfer to the theatre and the building of a separate maternity theatre."

- Correct spelling of "Caesarean"

- Was implementation of these interventions monitored or audited?

Did the training have the desired impact of improving fetal heart monitoring, neonatal resus, etc? Did delay in performing caesareans reduce?

P12 first paragraph: the reason for low coverage with surfactant is the cost. Could the authors explain how this was funded?

In your response to comments you have stated "Surfactant is a product that is paid for in our

hospital just like other commodities like antibiotics or caesarean section. The cost of surfactant is less than an operation like caesarean section.

Majority (98%) of the preterms that needed it were able to pay for it and actually received it

This is a private not for profit hospital, majority of the patients that are admitted are able to afford the rates offered by the hospital. We also get low cost surfactant from south Africa / India which costs about 200 dollars per vial."

This information needs to be included in the text of the article.

P12 lines 16-17: "The change may be attributable to improvement in resuscitation practices, following training and provision of equipment." – it may be argued that the need for neonatal resus training could have been predicted from cause of death and training could have been done without having to do perinatal death audits.

Did the audits contribute to the training, or to monitoring how well the training was put into practice?

P12 final paragraph – this is more or less repeating what was written in the previous section about prematurity. I recommend removing this paragraph and adding any important information into the previous section on prematurity.

P13: "This study suggests that a significant reduction in case fatality rate for intrapartum-related hypoxia can be expected to occur after the implementation of multiple interventions to improve the inpatient care of small and sick newborns." – I think we know that already. The question for this study is – did the audit make any additional impact? Or could you just have implemented those interventions already reported in the literature? I.e. is it worth the time and effort to do an audit of the deaths?

P13 lines 17-18: "The fact that only 2.3% of early neonatal deaths were attributed to infection is a surprising finding. In another study, infections accounted for around 14% of early neonatal deaths." – I think you need to discuss the issue of case mix. Infection is probably commonest in the lowest socio-economic groups. Nsambya is a private hospital and therefore probably has very few patients from the lowest socio-economic groups. This is another issue to mention in the section on limitations and generalisability of your findings.

Your response to comments includes the statement that most cases of neonatal sepsis cause late neonatal deaths (after 1 week) and so would be missed by the audit. You should also mention this in the text of the article here.

P13 line 32: "This proportion was highest for macerated stillbirths (88% unknown)" – this begs the question as to whether it is worth auditing macerated stillbirths. Did anything useful come out of the audit of these cases? Otherwise could the authors recommend that future audits should focus only on fresh stillbirths and neonatal deaths, where causes and avoidable factors are easier to identify?

	Discussion final section on “Effectiveness of perinatal death audit.” – some of this repeats what is already stated in the introduction. No need to repeat the same thing twice. However it would be useful to discuss what were the factors that contributed to the success of this audit, particularly in reducing early neonatal deaths, and why it did NOT appear to have an impact on stillbirths or perinatal deaths overall. That is a finding which needs to be discussed and has not really been addressed in the discussion at all. Do you believe this finding? Or do you think it could be because of the problem of only having 2 years of pre-intervention data, or insufficient numbers to reach statistical significance, or an increase in reporting deaths after the audit started? If you believe this was real, what could be the explanation? P14 line 41, spelling of “healthcre” needs to be corrected to “healthcare” The conclusion needs to be strengthened. It is rather generic at present and doesn’t add much to what is already know. The conclusion needs to state whether you can conclude that perinatal death audit reduces early neonatal deaths, but not stillbirths, or whether you think further research is needed to confirm or refute this finding. Also what the implications are for whether it is worth auditing stillbirths or whether audits should focus on early neonatal deaths only (which one could conclude from reading the abstract).
--	---

REVIEWER	Gabriela Vazquez Benitez HealthPartners Institute, USA
REVIEW RETURNED	26-Jul-2019

GENERAL COMMENTS	This is an important paper describing the result of a quality improvement initiative to improve perinatal outcomes and test whether the implementation of a death audit in addition to other quality improvement actions resulted in improved perinatal outcomes. The authors made improvements from previous submission by incorporating an interrupted time series analysis. Though the work presented here is important and well thought, it is limited by poorly organized presentation, poor description of methods and results. Some concerns are listed below. However, this list is not exhaustive.  - Acronyms should be spell out the first time they appear in the text. Two examples, IRR and KMC. Acronyms used in tables and figures need to be described in footnotes. - Exclude 0.05 from the statistical significance range (p-value <0.05) - Indicate when additional actions/quality improvements took place. Did they started at the same time as the death audit? Did a washout period was considered, for the time the death audit initiated while no other action was implemented? - Indicate the time scale. Are event rates for analysis done by year? When estimating the IRR correspond to a year change? - If data are presented by year, then the ‘contrafactual is estimated using only 2 time points which is not sufficient for a robust estimate. - The interrupted time series is poorly described. It is not clear what they men by the following statement ‘describe various modifications to the model which can be considered’. It is not clear what are the parameters in the model. It seems that they included and intercept term, a time term, and a period term (pre/intervention), but not a
---

	period times time term (interaction). Did they counted events or they offset by the number of births (or corresponding denominator). It is not clear why they are presenting the results without over-dispersion when they found that the count data required the over-dispersion parameter. Indicate that the model used corresponded to a Poisson model, or otherwise.  - Figure 2 can be omitted, y-axis needs to be included, labels need to be formatted, and acronyms need to be spelled out. - Indicate whether the study was powered to test for improvements in perinatal outcomes. Note that the power requires more time points, and only two time points are available in the pre-intervention period. This needs to be addressed as a limitation, or further retrospective data collection is needed for years before 2006. - To respond to several of previous concerns raised by other reviewers, describe whether characteristics of the population differed in the pre and post intervention period. - Did standard of care distribution change between pre-intervention and intervention period, or was only assessed during the intervention period? - Number of decimals. Tables are well presented, but data in the main manuscript are presented with different length. Use the same number of digits in the table as in the text. - The text need further organization, especially the discussion section.
--	--

VERSION 2 – AUTHOR RESPONSE

Editorial Requests	Response
Reviewer: 2	
The authors have now addressed the majority of the comments, and have performed an ITS analysis, for which they are to be congratulated. I am not an expert statistician so it may be worth asking a statistician to check they are happy with the analysis, but it looks fine to me.	Thank you.
Firstly, the article does not address all of the points on the Cochrane checklist for assessing quality of ITS studies:  - was the intervention independent of other changes? - was the shape of the intervention effect prespecified? - was the intervention unlikely to affect data collection? - was knowledge of the allocated interventions adequately prevented during the study? - were incomplete outcome data adequately addressed? - was the study free from selective outcome 	Yes, at the time of implementation of the audit, there were no other studies or interventions going No Not until I was asked to perform the ITS; the 'shape' used was linear (standard) No, data on mortality, case fatalities was routinely collected HMIS data from the register that was analysed. It would not have been possible to prevent staff from knowing that audits had been introduced. This knowledge could have impacted vigilance to record data accurately. I'm not sure how incomplete outcome data would be defined – data were derived from HMIS, so

reporting? - was the study free from other risks of bias?	there could be errors in counting (perhaps more so prior to the intervention) but there is no way we would be able to know this. Yes, because the outcome measure was the routinely collected data; not sure – see comments above Yes, perhaps (it's generally hard to assert freeness from bias)
Secondly, the conclusions are somewhat surprising – that the audit had a massive impact on early neonatal death rates, but no effect on perinatal deaths overall or on stillbirths. The authors need to discuss whether they believe this finding, or whether it could be due to the fact that there were only 2 time points before the introduction of the intervention (and so some random variation could have accounted for the “before” changes – Cochrane specifies that there should be at least 3 timepoints before the intervention); or limited numbers. What further research would they recommend? Would they recommend that guidelines are changed so that audits focus only on early neonatal deaths, as audits of stillbirths appear to have had no impact?	Thank you for drawing our attention to this. The number of perinatal deaths is defined by the numbers of stillbirth and early neonatal death. The high numbers of stillbirth inflated the numbers of perinatal death. Thus, it wasn't surprising that even though the audit made a difference to early neonatal deaths, this difference is neutralised when combined with stillbirths (perinatal deaths). This explanation has now been added to the discussion, under “Main findings”.
The discussion rightly mentions many of the interventions implemented – however a sceptic could read this and conclude that actually the audit was not needed because most of the interventions could have been predicted purely from cause of death (eg from the fact that hypoxia is the main cause, one could conclude that staff need training on use of partograms, and neonatal resus). The authors need to give more information on whether they think the audit itself added to the impact, or whether they could have achieved the same impact just by implementing these training interventions, without doing the audit.	The audit allowed to explore more in depth the causes of death, which contributed to a much better-defined actions to address the causes of death. We specified this in the “the audit process section”.
The article reads well and is almost ready for publication but there are a number of typos which need to be corrected (see below) and the font on the pie charts is still too small to be legible.	Thank you. Errors addressed.
Conclusion – correct spelling of “introduction” Article summary, last line: “The intervention did not had an effect” – should be “The intervention did not have an effect”	Corrected. Thank you.

P5 line 34 “numbers of death,” should be “numbers of deaths,” or “due to the large number of deaths”	Corrected. Thank you
P6 top of page – explain which types of professional were in the audit team. Was it just neonatologists / neonatal nurses and paediatricians, or did it also include obstetricians / midwives? Was the quality of care just regarding the paediatric quality of care, or was the quality of obstetric care / partogram use also considered?	This is mentioned in the methods. See the first sentence under “Data collection”. See also the section on “Intervention: the audit process”
P6 line 31 correct spelling of “Caesarean”	Corrected.
P6 last line “describe various modifications to the model which can be considered.” – this sounds like an instruction or comment which has been left in by mistake? Please check.	Deleted. Thank you.
P8 last line “Among the different types of mortalities, sub-optimal care was most common among early neonatal deaths the most (11.8%)” – delete “the most” at the end	Deleted.
P10 line 28 correct spelling of “intervention”	Corrected. Thank you.
P10 line 49 “This study assessed then introduction of pernatal audits” – correct to “This study assessed the introduction of perinatal audits”	Corrected. Thank you.
P10 line 54 correct spelling of “moreover”	Corrected. Thank you.
Strengths and limitations section should mention the fact that a major limitation is that there were only 2 timepoints of data BEFORE introduction of the audits. Cochrane standards require there to be at least 3 timepoints before the intervention starts. The problem is that if there was some random variation this could lead to misinterpretation of the results. So: it is possible that actually there WAS an effect of the audits on overall perinatal mortality, but there was a random decrease between 2006 and 2007. Also, it is possible that the effect on early neonatal deaths may not have been as pronounced as suggested by the analysis, because the increase between 2006 and 2007 may have been in part due to random variation from year to year. So we should be cautious in interpreting the results.	We agree with these important limitations and have added them in the abstract and the discussion (see “strength and limitations section”)
Also – it is worth mentioning that when audits are first introduced, very often the mortality rate appears to increase in the first year or two, simply because of improved recording of deaths. This has been shown in numerous studies on perinatal death	We agree with this point. A literature review to assess the effect of perinatal mortality audit found that in a study in Bangladesh, PMR and SBR increased initially due to an increase in improved information capture (See Pattinson R, Kerber K,

audits.	Waiswa P, et al. Perinatal mortality audit: counting, accountability, and overcoming challenges in scaling up in low- and middle-income countries. International journal of gynaecology and obstetrics: the official organ of the International Federation of Gynaecology and Obstetrics. 2009 Oct;107 Suppl 1:S113-21, s21-2). This point has been added to the limitations (under discussion).
Another possible limitation is the lack of external input to the audit process. The methods section does not mention who was on the audit team, but presumably it was only staff from the hospital? It is to be expected that doctors, nurses and midwives often find it difficult to be self-critical and so may not have reported as many problems with quality of care as may have been identified by an external expert.	Only the staff of Nsambya participated in the audit. This is mentioned in the methods. See the sections on “Data collection” and “Intervention: the audit process”.
P11 line 22: “poor neonatal resuscitation skills, and, poor management of preterm babies”- remove commas – should be “poor neonatal resuscitation skills and poor management of preterm babies”	Thank you for pointing this out. We made the correction in the manuscript.
P11 lines 31-32: “The interventions for improvement included: training in the use of the partograph, fetal heart monitoring, neonatal resuscitation, measures to reduce delay in performing a cesarean section by involving doctors in the timely preparation of mothers before transfer to the theatre and the building of a separate maternity theatre.”	The implementation of these interventions was monitored by weekly audits done. The maternity theatre was only built after 2014.
 - Correct spelling of “Caesarean” - Was implementation of these interventions monitored or audited? Did the training have the desired impact of improving fetal heart monitoring, neonatal resus, etc? Did delay in performing caesareans reduce? 	Spelling corrected. Yes, there was a reduction in delay in performing Caesarean sections. Additional explanation added in the manuscript. Other parameters were not recorded.
P12 first paragraph: the reason for low coverage with surfactant is the cost. Could the authors explain how this was funded? In your response to comments you have stated “Surfactant is a product that is paid for in our hospital just like other commodities like antibiotics or caesarean section. The cost of surfactant is less than an operation like caesarean section. Majority (98%) of the preterms that needed it were	This has been included in the text. See the section on “Interventions for prematurity-related complications” (Discussion).

able to pay for it and actually received it This is a private not for profit hospital, majority of the patients that are admitted are able to afford the rates offered by the hospital. We also get low cost surfactant from south Africa / India which costs about 200 dollars per vial.” This information needs to be included in the text of the article.	
P12 lines 16-17: “The change may be attributable to improvement in resuscitation practices, following training and provision of equipment.” – it may be argued that the need for neonatal resus training could have been predicted from cause of death and training could have been done without having to do perinatal death audits. Did the audits contribute to the training, or to monitoring how well the training was put into practice?	The audits assisted to identify the gaps in knowledge and skill of resuscitation. During the audit a review of the charts was done, poor resuscitation skills were identified. Because any baby with a low Apgar score the steps in resuscitation were documented and many times especially at the beginning there was missing key components. Audit was used to assist to monitor the frequency of the resuscitations done. Therefore, it was very useful.
P12 final paragraph – this is more or less repeating what was written in the previous section about prematurity. I recommend removing this paragraph and adding any important information into the previous section on prematurity.	Difficult to identify the referred section from page reference, but we agree in principle to remove any repetitive text.
P13: “This study suggests that a significant reduction in case fatality rate for intrapartum-related hypoxia can be expected to occur after the implementation of multiple interventions to improve the inpatient care of small and sick newborns.” – I think we know that already. The question for this study is – did the audit make any additional impact? Or could you just have implemented those interventions already reported in the literature? I.e. is it worth the time and effort to do an audit of the deaths?	The ITS analysis now included was meant to identify any difference in outcomes. Without the audit it is difficult to identify which actions to take and make efficient use of mega resources. This explanation is included in the manuscript.
P13 lines 17-18: “The fact that only 2.3% of early neonatal deaths were attributed to infection is a surprising finding. In another study, infections accounted for around 14% of early neonatal deaths.” – I think you need to discuss the issue of case mix. Infection is probably commonest in the lowest socio-economic groups. Nsambya is a private hospital and therefore probably has very few patients from the lowest socio-economic groups. This is another issue to mention in the section on limitations and generalisability of your findings. Your response to comments includes the statement that most cases of neonatal sepsis cause late neonatal deaths (after 1 week) and so would be	This has been included in the text.

missed by the audit. You should also mention this in the text of the article here.	
P13 line 32: “This proportion was highest for macerated stillbirths (88% unknown)” – this begs the question as to whether it is worth auditing macerated stillbirths. Did anything useful come out of the audit of these cases? Otherwise could the authors recommend that future audits should focus only on fresh stillbirths and neonatal deaths, where causes and avoidable factors are easier to identify?	The high proportion of cases with unknown cause of death among macerated stillbirth is not a new finding. In low-resource settings, where pregnancy monitoring is suboptimal, less information is available for macerated stillbirth. Other studies have reported similar results as well. The WHO perinatal death audit guidelines (2016) recommend focusing more on intrapartum (“fresh”) stillbirths and neonatal deaths. However, this is not because of high proportion of cases with unknown cause, but because it is easier to prevent intrapartum stillbirths and neonatal deaths than antepartum (“macerated”) stillbirth – the so-called low-hanging fruits.
Discussion final section on “Effectiveness of perinatal death audit.” – some of this repeats what is already stated in the introduction. No need to repeat the same thing twice. However it would be useful to discuss what were the factors that contributed to the success of this audit, particularly in reducing early neonatal deaths, and why it did NOT appear to have an impact on stillbirths or perinatal deaths overall. That is a finding which needs to be discussed and has not really been addressed in the discussion at all. Do you believe this finding? Or do you think it could be because of the problem of only having 2 years of pre-intervention data, or insufficient numbers to reach statistical significance, or an increase in reporting deaths after the audit started? If you believe this was real, what could be the explanation?	We agree that it was repetitive. We’ve removed the repetitions in the revised manuscript. We believe that the limitation of having only two time points pre intervention might explain the finding. We added this in the “limitations” section of the discussion, and suggested that other ITS studies with longer pre-intervention timepoints of data collection in LMICs should confirm or refute our findings, in the “Effectiveness of perinatal death audit” section
P14 line 41, spelling of “healtchre” needs to be corrected to “healthcare”	Corrected. Thank you.
The conclusion needs to be strengthened. It is rather generic at present and doesn’t add much to what is already know. The conclusion needs to state whether you can conclude that perinatal death audit reduces early neonatal deaths, but not stillbirths, or whether you think further research is needed to confirm or refute this finding. Also what the implications are for whether it is worth auditing stillbirths or whether audits should focus on early neonatal deaths only (which one could conclude from reading the abstract).	We agree, we added that further research is needed to confirm or refute findings in the “conclusion”. Moreover, we added in the “Effectiveness of perinatal death audit” section that post mortem autopsies should be included in studies assessing the effect of perinatal audits on stillbirths, as without this, assigning causes of stillbirth can be challenging.

Reviewer: 5 (stats review)	
This is an important paper describing the result of a quality improvement initiative to improve perinatal outcomes and test whether the implementation of a death audit in addition to other quality improvement actions resulted in improved perinatal outcomes. The authors made improvements from previous submission by incorporating an interrupted time series analysis.	Thank you.
Though the work presented here is important and well thought, it is limited by poorly organized presentation, poor description of methods and results.	Improvements made throughout. Specific comments have been addressed or responded to.
Some concerns are listed below. However, this list is not exhaustive. - Acronyms should be spell out the first time they appear in the text. Two examples, IRR and KMC. Acronyms used in tables and figures need to be described in footnotes.	Thank you. Acronyms defined.
- Exclude 0.05 from the statistical significance range (p-value <0.05)	Corrected. Thank you.
- Indicate when additional actions/quality improvements took place. Did they start at the same time as the death audit? Did a washout period was considered, for the time the death audit initiated while no other action was implemented?	The action points took place simultaneously with the Audit. Audit was done every week and once a gap was identified, we a recommendation was made and this was implemented and reviewed during the subsequent meetings. This is detailed in the discussion under "Quality improvement".
- Indicate the time scale. Are event rates for analysis done by year? When estimating the IRR correspond to a year change?	Yes, time scale is year. Amendments made in the results.
- If data are presented by year, then the 'contrafactual is estimated using only 2 time points which is not sufficient for a robust estimate.	Yes, we agree. This has been highlighted as a limitation.
- The interrupted time series is poorly described. It is not clear what they men by the following statement 'describe various modifications to the model which can be considered'. It is not clear what are the parameters in the model. It seems that they included and intercept term, a time term, and a period term (pre/ intervention), but not a period times time term (interaction). Did they counted events or they offset by the number of births (or corresponding denominator).	This has now been amended. The data available were for 10 years, two of which were prior to the intervention. Since a level change was hypothesised the inclusion of a period by time interaction would over-fit the data prior to intervention and was therefore not considered. Thank you for highlighting this omission which has

It is not clear why they are presenting the results without over-dispersion when they found that the count data required the over-dispersion parameter. Indicate that the model used corresponded to a Poisson model, or otherwise.	now been clarified in the methods. Results with over-dispersion considered should have greater weight in reporting (indeed those without over-dispersion could be dropped). Now stated in methods.
- Figure 2 can be omitted, y-axis needs to be included, labels need to be formatted, and acronyms need to be spelled out.	We agree. We've omitted Fig 2.
- Indicate whether the study was powered to test for improvements in perinatal outcomes. Note that the power requires more time points, and only two time points are available in the pre-intervention period. This needs to be addressed as a limitation, or further retrospective data collection is needed for years before 2006.	Results are provided with confidence intervals which indicate the precision of estimates, which reflects the power.
- To respond to several of previous concerns raised by other reviewers, describe whether characteristics of the population differed in the pre and post intervention period.	Audit was not done before. Thus, the reported outcomes were the only data points before the study.
- Did standard of care distribution change between pre-intervention and intervention period, or was only assessed during the intervention period?	Before the intervention, there was no regular assessment of standard of care.
- Number of decimals. Tables are well presented, but data in the main manuscript are presented with different length. Use the same number of digits in the table as in the text.	Thank you. All identified errors now corrected.
- The text need further organization, especially the discussion section.	Discussion re-organised. Thank you.

VERSION 3 – REVIEW

REVIEWER	Merlin Willcox School of Primary Care, Population Sciences and Medical Education University of Southampton UK I am collaborating with Dr Victoria Nakibuuka and Dr Mamuda Aminu in writing a grant proposal. However I have not allowed this to influence my judgement on this article.
REVIEW RETURNED	01-Oct-2019

GENERAL COMMENTS	NB the expert statistical review has already been done, but it would
--

be good to check that the statistical reviewer is happy with the changes made.

Thank you to the authors for responding to our comments - I think the article is almost ready for publication but there remain a few issues which could be better addressed.

1. Thank you for responding to the Cochrane questions on risk of bias in an ITS study. This information needs to be incorporated into the methods section of the article, so that anyone doing a systematic review can complete the risk of bias assessment.

There is also one point which may need to be nuanced. You state that "at the time of implementation of the audit there were no other interventions going on". Are you very sure? What about implementation of maternal death reviews – when was this introduced in the hospital? This could also have had an impact. In the QUARITE trial, Dumont et al (2013) showed that simply conducting maternal death reviews resulted in a 25% reduction in early neonatal mortality (but no reduction in stillbirths) – similar to your study. Construction of a new maternity theatre just one year into the reviews seems like a major intervention (however please clarify the time in the text – currently the discussion implies that the theatre was opened in 2008, but in the response to reviewers you say it opened in 2014). Was this honestly only the result of perinatal death reviews (this seems somewhat unlikely to me!), or had it been planned anyway, or maybe it could have been recommended by maternal death reviews also? The point is that there probably are some concurrent interventions which could also have impacted perinatal mortality, and these need to be recognised and discussed. One might also expect there to be a general improvement in mortality over time, however you could quote data from the Uganda DHS surveys which show that, at least at the national level, perinatal mortality did not improve over this time period.

2. Thank you for adding the explanation to "main findings" that "Thus, even though the audit made a difference to early neonatal deaths, this difference is neutralised when combined with stillbirths." This is true however it does not address my main concern, which is about the main conclusion of your study as stated in the abstract: "The introduction of perinatal death audit showed no statistically significant effect on perinatal mortality or stillbirth rate, but a significant decrease in early neonatal mortality rate." This is a somewhat surprising finding to those promoting perinatal death audits, and deserves to be discussed in the discussion. Currently this is not adequately discussed. We need a section in the discussion about "why was there no reduction in the stillbirth rate in this study?". Either (a) The implication of this conclusion is that we should stop auditing stillbirths and ONLY focus on early neonatal deaths. Is this what you want to say? If so, you perhaps need to back this up by reference to other studies such as Dumont et al (2013) mentioned above.

Or (b) there could be other reasons why there was no impact on stillbirth rates.

Did you try to separate out the effect on fresh and macerated stillbirths? One would not expect perinatal death audit to have much impact on the incidence of macerated stillbirths (unless some are misclassified fresh stillbirths), because the avoidable factors (if any) would usually have occurred well before the mother arrived in the hospital. However intrapartum asphyxia is likely to be the major cause of fresh stillbirths (in your supp table 2 it is almost 50% of the fresh stillbirths) as well as early neonatal deaths, so logically one would expect some effect in reducing fresh stillbirths. If there really

was no effect even in fresh stillbirths, reasons for this need to be discussed in the discussion. One alternative explanation is that this is not actually a true finding, by random occurrence there may have been a “blip” in stillbirth rates in the “before” time points, which gives the false impression that the audit had no effect on stillbirths. Hence the need for caution in interpreting this finding. Or if it is really true that there was no change in fresh stillbirths, was it because obstetricians were not sufficiently involved in the perinatal death audits? However your methods state that obstetricians and midwives were involved. In your QI section you mention several interventions which should have reduced the rate of fresh stillbirths: constructing a new theatre, to reduce decision-to-incision time for CS; training and equipment for neonatal resus; and training midwives to use partograms better. You state in your response to reviewers that “implementation of recommendations was monitored by weekly audit” and that Decision to incision time was reduced – but I cannot see where this is stated in the main article. Since there was no criterion-based clinical audit to see whether process indicators improved, we do not know whether neonatal resus was done better. Was neonatal resus done even on fresh stillbirths? In some countries this is now routine, because it may be possible to successfully resuscitate some babies who are labelled initially as a “fresh stillbirth”. Were partograms used better after the training? (if this was not audited, I suppose we don’t know). It is possible, for example, that since the lead author is a neonatologist, more emphasis was given to high quality neonatal resus (explaining the reduction in early neonatal deaths) and less on high quality use of the partogram (explaining the lack in change of stillbirths). Training of midwives to use the partogram is not necessarily sufficient to lead to improved use. Or could it be because many fresh stillbirths resulted from delayed referrals to the hospital from other facilities? Again if this was the case, it needs to be included in the discussion. From having been involved in the whole process, the authors must have some feeling for why there appeared to be no effect on stillbirths, and if anything could have been done differently to improve the impact on stillbirths. Even if this is only anecdotal, the authors’ experiences will be very informative for others wanting to replicate and improve upon this perinatal death audit process.

3. The font in the pie charts is still too small to be legible (fig 1). These don’t seem to have been changed since the previous version.

4. In your response to reviewers, you state that you have added to the discussion (limitations) section something about death rates appearing to increase after introduction of audit, because of better reporting. However I cannot see this in the discussion section anywhere.

5. The lack of external input into the audit process is mentioned in the methods but it is not mentioned as a possible limitation. This could be another explanation for the lack of impact on stillbirths.

6. Your explanation in the response to reviewers about how the detailed perinatal death audit helped to rectify errors in the neonatal resuscitation process needs to be added to the main text of the article: “The audits assisted to identify the gaps in knowledge and skill of resuscitation. During the audit a review of the charts was done, poor resuscitation skills were identified. Because any baby with a low Apgar score the steps in resuscitation were documented and many times especially at the beginning there was missing key components. Audit was used to assist to monitor the frequency of the resuscitations done.”

	7. Removing duplication: I was suggesting that the paragraph starting “The case fatality rate for prematurity was reduced by more than half...” – now on p14 of the current manuscript – should be deleted and any important information added to the section entitled “Interventions for prematurity-related complications” – otherwise there is some duplication. 8. Please add to your discussion the statement from response to reviewers that “The WHO perinatal death audit guidelines (2016) recommend focusing more on intrapartum (“fresh”) stillbirths and neonatal deaths.” 9. Recommendation to add postmortem autopsies: Is this likely to be culturally acceptable? How would this be funded? What extra useful information would it yield? Even in the UK it is often hard to establish cause of many IUFDs. Wouldn’t it be better to focus resources on improving implementation of existing recommendations? 10. As suggested by the statistical reviewer, it might be clearer to present only the results taking into account “over-dispersion” if this was felt to be the best method.
--	---

VERSION 3 – AUTHOR RESPONSE

Comment	Response
Reviewer 2	
NB the expert statistical review has already been done, but it would be good to check that the statistical reviewer is happy with the changes made.	All statistical changes were made under the direct guidance of our Senior Medical Statistician.
Thank you to the authors for responding to our comments - I think the article is almost ready for publication but there remain a few issues which could be better addressed.	Thank you.
1. Thank you for responding to the Cochrane questions on risk of bias in an ITS study. This information needs to be incorporated into the methods section of the article, so that anyone doing a systematic review can complete the risk of bias assessment.	This is now indicated in the methods.
There is also one point which may need to be nuanced. You state that “at the time of implementation of the audit there were no other interventions going on”. Are you very sure? What about implementation of maternal death reviews – when was this introduced in the hospital? This could also have had an impact. In the QUARITE trial, Dumont et al (2013) showed that simply conducting maternal death reviews resulted in a 25% reduction in early neonatal mortality (but no reduction in stillbirths) – similar to your study.	Yes, maternal death reviews were done. But there was no interaction between paediatricians and the obstetricians. The focus was mainly on mothers.
Construction of a new maternity theatre just one year into the reviews seems like a major intervention (however please clarify the time in the text – currently the discussion implies	This was an oversight and thank you for highlighting it. A recommendation to build the maternity was done in 2008. However, it was opened in 2014. This is now corrected in the main

that the theatre was opened in 2008, but in the response to reviewers you say it opened in 2014). Was this honestly only the result of perinatal death reviews (this seems somewhat unlikely to me!), or had it been planned anyway, or maybe it could have been recommended by maternal death reviews also? The point is that there probably are some concurrent interventions which could also have impacted perinatal mortality, and these need to be recognised and discussed. One might also expect there to be a general improvement in mortality over time, however you could quote data from the Uganda DHS surveys which show that, at least at the national level, perinatal mortality did not improve over this time period.	document. It is difficult to ascertain it was due to maternal death reviews. However, during the perinatal death meetings especially for fresh stillbirths, it was clear that many cases took more than 1 hour for a caesarean section to be done as they were waiting for theatre space. Several meetings were held with the administration to see this come through.
2. Thank you for adding the explanation to “main findings” that “Thus, even though the audit made a difference to early neonatal deaths, this difference is neutralised when combined with stillbirths.” This is true however it does not address my main concern, which is about the main conclusion of your study as stated in the abstract: “The introduction of perinatal death audit showed no statistically significant effect on perinatal mortality or stillbirth rate, but a significant decrease in early neonatal mortality rate.” This is a somewhat surprising finding to those promoting perinatal death audits, and deserves to be discussed in the discussion. Currently this is not adequately discussed. We need a section in the discussion about “why was there no reduction in the stillbirth rate in this study?”. Either (a) The implication of this conclusion is that we should stop auditing stillbirths and ONLY focus on early neonatal deaths. Is this what you want to say? If so, you perhaps need to back this up by reference to other studies such as Dumont et al (2013) mentioned above. Or (b) there could be other reasons why there was no impact on stillbirth rates.	We agree with this comment. However, we think that our findings do not allow stating that stillbirths cases should not be audited any longer. Thus, we decided to put our results in perspective, comparing them more with other publications. Besides, this finding was not a surprise to us considering that contemporaneous evidence shows that 50% of stillbirths in similar settings are antepartum deaths (Aminu et al, 2019), the majority of which occur outside the hospital. Thus, it is expected that improvements in hospital care may not have as much effect on stillbirths (since a large proportion happens at home) as it would on newborn outcomes. We’ve also mentioned that “While it is difficult to attribute all changes in perinatal outcomes to perinatal death audit alone, the process has helped to identify areas of healthcare that need support for improvement in outcomes”. (see “Strengths and Limitations”). Hope this is clearer. The results have now been compared with those of Dumont et al (see “Effectiveness of perinatal death audit”). Thank you.
Did you try to separate out the effect on fresh and macerated stillbirths? One would not expect perinatal death audit to have much impact on the incidence of macerated stillbirths (unless some are misclassified fresh stillbirths), because the avoidable factors (if any) would usually have occurred	Thank you for this comment. It is common to assume that “fresh” stillbirths are intrapartum deaths and “macerated” stillbirths are antepartum. While this may be true most of the time in high-income settings, where there is adequate monitoring of both pregnancy and labour, it is frequently misleading in low-resource settings like

well before the mother arrived in the hospital. However intrapartum asphyxia is likely to be the major cause of fresh stillbirths (in your supp table 2 it is almost 50% of the fresh stillbirths) as well as early neonatal deaths, so logically one would expect some effect in reducing fresh stillbirths. If there really was no effect even in fresh stillbirths, reasons for this need to be discussed in the discussion.	our study setting. In LMIC, women frequently start labour with a live baby and spend 24 hours or more in labour. Since maceration starts within 4-6 hours after death (Genest & Singer, 1992), many of the so-called macerated stillbirths are actually deaths during labour and childbirth (intrapartum). There is also evidence that healthcare workers in low-resource settings misclassify up to one-third of “fresh” stillbirths as “macerated” (Gold et al, 2014). Therefore, we do not use the assumption that “fresh SB” equals intrapartum death and “macerated SB” equals antepartum death. (Please see also Aminu et al (2019) for a more pragmatic approach to determining the time of death in low-resource settings).
One alternative explanation is that this is not actually a true finding, by random occurrence there may have been a “blip” in stillbirth rates in the “before” time points, which gives the false impression that the audit had no effect on stillbirths. Hence the need for caution in interpreting this finding. Or if it is really true that there was no change in fresh stillbirths, was it because obstetricians were not sufficiently involved in the perinatal death audits?	Please see previous responses for the explanation on the lack of impact on stillbirth from our perspective and as far as one could infer from the data within reason. We had an equal number of paediatricians and obstetricians in the audit.
In some countries this is now routine, because it may be possible to successfully resuscitate some babies who are labelled initially as a “fresh stillbirth”. Were partograms used better after the training? (if this was not audited, I suppose we don’t know). It is possible, for example, that since the lead author is a neonatologist, more emphasis was given to high quality neonatal resus (explaining the reduction in early neonatal deaths) and less on high quality use of the partogram (explaining the lack in change of stillbirths).	We implemented the interventions as a multidisciplinary team, not as an individual. It is unlikely that one neonatologist can influence how the other group of 50 people can implement changes.
Training of midwives to use the partogram is not necessarily sufficient to lead to improved use. Or could it be because many fresh stillbirths resulted from delayed referrals to the hospital from other facilities? Again if this was the case, it needs to be included in the discussion. From having been involved in the whole process, the authors must have some feeling for why there appeared to be no effect on stillbirths, and if anything could have been done differently to improve the impact on stillbirths. Even if this is only anecdotal, the	From our data, about 17% of the cases were referrals (see Table 1 in the manuscript). There is evidence that training could improve the retention of both knowledge and skills necessary to improve practice (Ameh et al, 2018).

authors' experiences will be very informative for others wanting to replicate and improve upon this perinatal death audit process.	
However your methods state that obstetricians and midwives were involved. In your QI section you mention several interventions which should have reduced the rate of fresh stillbirths: constructing a new theatre, to reduce decision-to-incision time for CS; training and equipment for neonatal resus; and training midwives to use partograms better. You state in your response to reviewers that "implementation of recommendations was monitored by weekly audit" and that Decision to incision time was reduced – but I cannot see where this is stated in the main article. Since there was no criterion-based clinical audit to see whether process indicators improved, we do not know whether neonatal resus was done better. Was neonatal resus done even on fresh stillbirths?	Neonatal resuscitation was not done for fresh stillbirths. The healthcare workers were competent enough to know that a stillborn baby cannot be resuscitated. Improvement in neonatal resuscitation was noted during the perinatal death audit meetings when we reviewed the documentation of the type resuscitation done for babies with low Apgar scores. The steps in resuscitations were clearly followed and things like Aminophylline and dextrose were not given as before.
3. The font in the pie charts is still too small to be legible (fig 1). These don't seem to have been changed since the previous version.	The fonts were increased to size 10, which we thought was reasonably large in a figure. However, we've now increased the font size to 12 and increased the overall size of the figure to accommodate the larger fonts.
4. In your response to reviewers, you state that you have added to the discussion (limitations) section something about death rates appearing to increase after introduction of audit, because of better reporting. However I cannot see this in the discussion section anywhere.	This was the view of a reviewer, but it may be a possibility.
5. The lack of external input into the audit process is mentioned in the methods but it is not mentioned as a possible limitation. This could be another explanation for the lack of impact on stillbirths.	Thank you, but we disagree with this point because the main purpose of external reviewers in an audit team is to ensure unbiased identification of causes and factors contributing to the deaths. They have very limited input (if any) on the interventions that follow audit. Please see previous responses for a possible explanation, from our perspective, for the lack of impact on stillbirths.
6. Your explanation in the response to reviewers about how the detailed perinatal death audit helped to rectify errors in the neonatal resuscitation process needs to be added to the main text of the article: "The audits assisted to identify the gaps in knowledge and skill of resuscitation. During the audit a review of the charts was done, poor resuscitation skills were identified.	This was in response to a specific comment. However, we feel that this level of detail may not be appropriate in the manuscript. We believe that it is perfectly reasonable to assume that audit would help identify gaps in care and guide improvement in practice since that is the primary purpose of the audit.

Because any baby with a low Apgar score the steps in resuscitation were documented and many times especially at the beginning there was missing key components. Audit was used to assist to monitor the frequency of the resuscitations done.”	
7. Removing duplication: I was suggesting that the paragraph starting “The case fatality rate for prematurity was reduced by more than half...” – now on p14 of the current manuscript – should be deleted and any important information added to the section entitled “Interventions for prematurity-related complications” – otherwise there is some duplication.	We feel this is not a duplication because even though the two sections discuss issues related to prematurity, the issues are different. We believe that the reader should be reminded of the results before points are discussed.
8. Please add to your discussion the statement from response to reviewers that “The WHO perinatal death audit guidelines (2016) recommend focusing more on intrapartum (“fresh”) stillbirths and neonatal deaths.”	Thank you. The statement was in response to a specific comment. It is unclear where it would seem relevant to the points we are making.
9. Recommendation to add postmortem autopsies: Is this likely to be culturally acceptable? How would this be funded? What extra useful information would it yield? Even in the UK it is often hard to establish cause of many IUFDs. Wouldn't it be better to focus resources on improving implementation of existing recommendations?	While we've made other recommendations, it is important to note that autopsy is still the gold standard for the identification of the cause of perinatal death.
10. As suggested by the statistical reviewer, it might be clearer to present only the results taking into account “over-dispersion” if this was felt to be the best method.	Only results with over-dispersion are now presented. Thank you. (see Fig 2 and Table 2).

VERSION 4 – REVIEW

REVIEWER	Merlin Willcox School of Primary Care, Population Sciences and Medical Education University of Southampton UK I am collaborating with Dr Victoria Nakibuuka and Dr Mamuda Aminu in writing a grant proposal. However I have not allowed this to influence my judgement on this article.
REVIEW RETURNED	17-Dec-2019

GENERAL COMMENTS	Review of Nakibuuka et al, “Prospective Study to Explore Changes in Quality of Care and Perinatal Outcomes After Implementation of Perinatal Death Audit in Uganda”
---

(version 4, Nov 2019)

I apologise for the delay in providing this review, due to multiple competing priorities over the last month.

I have read the latest version of the article and the authors' responses to my comments. It is a shame that they have chosen not to incorporate many of the suggested changes into the manuscript. I note the remark that you normally only allow 2 revisions and this was already the third.

On balance I believe that this is important work and needs to be published, so I would not want my comments to prevent publication. However, I would like to make two remarks for consideration of the editor, who can decide whether or not to ask the authors to address these.

1. The discussion is still heavily focussed on explaining the positive interventions which had an impact on neonatal mortality. That is great – but there is still a lack of discussion to reflect upon and attempt to explain the lack of effect on stillbirths (which is an equally important finding). The response to my comments indicates that the authors have thought about this, but they have chosen not to put this into the discussion, which makes the article weaker, and leaves the reader with the impression that actually there is no hope for perinatal death audit to have any impact on stillbirths (which may not be correct – some other studies in similar low-income contexts HAVE shown a reduction in stillbirths – see for example Bugalho A, Bergstrom S. Value of perinatal audit in obstetric care in the developing world: a ten-year experience of the Maputo model. *Gynecologic & Obstetric Investigation*. 1993;36(4):239-43.; and Ward HR, Howarth GR, Jennings OJ, Pattinson RC. Audit incorporating avoidability and appropriate intervention can significantly decrease perinatal mortality. *South African Medical Journal*. 1995;85(3):147-50.). Why is it that some others have found a reduction in stillbirths whereas this study didn't? Even if 50% of stillbirths occur outside of health facilities (as stated by the authors), that implies that another 50% occurs in facilities and so could be addressed. The question is how the process here could be improved to have an impact on stillbirths, rather than simply stating that it didn't have an impact, and assuming that it cannot have an impact.

2. In response to my question as to whether neonatal resus was done even on apparent fresh stillbirths, the authors replied "Neonatal resuscitation was not done for fresh stillbirths. The healthcare workers were competent enough to know that a stillborn baby cannot be resuscitated." Actually in many countries guidelines state that any "flat baby" (including apparent fresh stillbirths) SHOULD be resuscitated and that many can be discharged alive and well. See for example: <https://fn.bmj.com/content/78/2/F112>

For example see the UK guidelines which state that any baby without detectable cardiac activity or breathing SHOULD be resuscitated and resuscitation should ONLY be abandoned if there is still no detectable cardiac activity after 10 minutes of resuscitation: <https://www.resus.org.uk/resuscitation-guidelines/resuscitation-and-support-of-transition-of-babies-at-birth/>

In conclusion I think this article should be published but I still maintain that the article and its readers would benefit from a deeper reflection on why there was no impact on stillbirths, and how the process could be further improved in order to impact on stillbirths in

	the future. It is up to the editor to decide whether to press this point. I have made suggestions which the authors have chosen not to include. The reviewer provided a marked copy with additional comments. Please contact the publisher for full details.
--	--

VERSION 4 – AUTHOR RESPONSE

1. Throughout the manuscript, we have tried to present a balanced view of our findings. Contrary to the reviewer’s claim, there isn’t a single conscious attempt to choose favourable discussion points. We believe we have adequately highlighted the limitations of our study both in the section for “Strengths and limitations of this study” and in the discussion.

The reviewer may have a more desirable outcome in mind, which we regret that we are unable to justify simply because our results do not support the same view.

To be clear, we are not disputing that there is some evidence that audit could decrease perinatal mortality. We have, in fact, cited some of the evidence in support of that view (Pattinson et al, 2009; Eskes, 2014).

However, we have also cited in our discussion (under the subheading “Effectiveness of perinatal death audit”) a study in South Africa where only 30% of 54 health facilities that started perinatal death audit reported a decrease in perinatal mortality, while 35% had an INCREASE in mortality (Allanson & Pattinson, 2015).

Therefore, while we favour the view that audit could reduce perinatal mortality, we must also recognise that the available evidence is still few and far between (often conflicting and methodologically weak) for one to take a conclusive position – at least not with the currently available evidence.

Regarding our findings, we must reiterate that perinatal mortality is defined as stillbirth AND early neonatal death. This means that a reduction in neonatal mortality may mean a reduction in perinatal mortality as well. The same applies to stillbirth. It also means that when an intervention reduces early neonatal mortality, it may reduce the overall perinatal mortality, but may not necessarily affect stillbirth. As a result, the impact of the intervention on the overall perinatal mortality could be neutralised when it affects one type of mortality and not the other.

Therefore, for the above reasons ((i) that not all studies find evidence that audit reduces perinatal mortality (ii) that interventions can affect neonatal death, but not stillbirth), it was not surprising for us that we found a reduction in neonatal death but not in stillbirth. Nonetheless, we believe our results add valuable data to this important discussion.

It is possible that our chosen interventions were more effective for neonatal death than for stillbirth. It is also possible that actions for neonatal death (such as resuscitation) were executed more effectively than those for stillbirth - since it is the actions that make the difference, not the mere conduct of audit. However, we cannot substantiate any of these explanations with any evidence.

2. The reviewer’s question was, “Was neonatal resus done even on fresh stillbirths?” There is a difference between a fresh stillbirth and a “flat baby”. While a “flat baby” is alive (hence, the recommendation to resuscitate), a stillbirth is (by definition) a DEAD baby and cannot be resuscitated. Had the reviewer made this distinction, it would have been much clearer. In our facility, a baby is only declared a stillborn when confirmed dead.